# SeerAttention: Self-distilled Attention Gating for Efficient Long-context Prefilling

**Yizhao Gao**[1][*][†]   **Zhichen Zeng**[2][*][†]   **Dayou Du**[3][†]   **Shijie Cao**[4][‡]   **Peiyuan Zhou**[5]
**Jiaxing Qi**[5]   **Junjie Lai**[5]   **Hayden Kwok-Hay So**[1]   **Ting Cao**[6]   **Fan Yang**[4]   **Mao Yang**[4]
[1]University of Hong Kong    [2]University of Washington
[3]University of Edinburgh    [4]Microsoft Research    [5]NVIDIA    [6]Tsinghua University

## Abstract

Attention is the cornerstone of modern Large Language Models (LLMs). Yet its quadratic complexity hinders efficiency and scalability, especially for long-context processing. A promising approach is to leverage sparsity in attention. However, existing sparsity-based solutions predominantly rely on *predefined patterns or heuristics* at the attention head level, struggling to adapt dynamically to different contexts efficiently. We propose SeerAttention, a simple yet effective attention mechanism that directly learns the block-level attention sparsity from the LLM itself. Inspired by the gating mechanism in Mixture of Experts (MoE), SeerAttention augments the conventional attention with a **learnable gate** that **selectively activates important blocks** within the attention map. Specifically, the gate first pools the query (Q) and key (K) tensors along the sequence dimension and processes them through learnable linear layers. The resulting matrices are then multiplied together to produce the gating scores, which are used to predict block-level attention sparsity. Combined with our block-sparse FlashAttention kernel, SeerAttention can achieve significant speedup on GPUs. When applied to pre-trained LLMs, SeerAttention only requires training the gate parameters in a lightweight self-distillation manner, allowing rapid convergence. Our evaluation results demonstrate that SeerAttention achieves better model accuracy and lower latency for long-context pre-filling compared to prior methods. Code is available at: https://github.com/microsoft/SeerAttention.

## 1 Introduction

Attention is a fundamental mechanism in transformer-based LLMs [51]. Despite its effectiveness, the quadratic complexity of attention demands substantial computation and memory resources, limiting the scalability and efficiency of LLMs, especially for long-context windows. This challenge has become an active research topic in the community. One potential solution is to replace the quadratic attention with cheaper architectures like linear attention or recurrent networks [30, 20, 40, 47] with subquadratic complexity. While these approaches are more efficient, the majority of state-of-the-art large language models (LLMs) continue to use full attention to achieve better performance.

A promising approach with increasing interests is to leverage sparsity in attention. Sparsity commonly exists in attention maps, and it becomes more prominent in longer contexts. In certain LLM attention heads, the sparsity ratio can reach 95% or even 99%, posing great opportunities for efficiency improvements. However, prior post-training methods often rely on predefined sparsity patterns or heuristics to approximate the attention mechanism [28, 18, 32, 64, 22, 54]. The sparsity observed in

---

[*]Equal Contribution.

[†]Work partially done during the internship at Microsoft Research.

[‡]Corresponding author: shijiecao@microsoft.com.

attention maps varies significantly across different models, input contexts and attention heads, making predefined patterns or heuristics insufficient. On the other hand, pre-training a sparse attention model from scratch like Native Sparse Attention [58] or MoBA [38] is costly and can not be directly applied to other dense pre-trained models.

In this paper, we introduce SeerAttention, a simple yet effective post-training distillation method that brings tangible attention sparsity to any full-attention models through self-distillation without relying on predefined sparsity patterns. To achieve this, SeerAttention augments conventional attention with a learnable gate called *AttnGate* that selectively activates a small subset of important blocks in the attention map, drawing inspiration from the gating mechanism in MoE [45]. The AttnGates are trained with the 2D block-level sparsity ground truth generated by the original LLMs. Importantly, this distillation process only requires learning the gating parameters, while all other model parameters remain fixed. This leads to fast training process as only the newly added gate weights requires to compute gradient. In this way, without relying on human heuristic observations, users can obtain tailored AttnGates for different models.

Our results demonstrate that SeerAttention surpasses state-of-the-art post-training sparse attention methods like MInference [28], MoA [18] and DuoAttention [54] in terms of long context model accuracy and pre-filling latency. SeerAttention achieves highly linear speedup over dense configurations, delivering a 7.3× speedup with 90% sparsity on sequences of 128k. Notably, in contrast to previous methods that require careful calibration of sparse configuration for different heads, SeerAttention offers strong capabilities of adaptation to different heads and contexts. Remarkably, on top of block-sparse pattern, SeerAttention exhibits the ability to learn more diverse patterns, including A-shape and Vertical-Slash, further demonstrating its versatility and performance.

Our contributions can be summarized as follows:

- We propose SeerAttention, an innovative learnable attention gating mechanism to enhance efficiency for long-context LLMs.

- We have developed a self-distillation training scheme to efficiently train the AttnGate, enabling it to learn the intrinsic sparsity of a pre-trained model.

- Experiments show that SeerAttention outperforms previous approaches, offering adaptability to various context lengths and sparsity ratios.

## 2  Background and Related Works

**Powerful but Complex Attention in Transformer.**    The advent of attention mechanisms, particularly within the Transformer architecture [51], marked a significant advancement in natural language processing. Attention enables improved handling of long-range dependencies and a better understanding of context by attending each token to every other token in the sequence, resulting in a quadratic memory and time complexity $O(n^2)$, where $n$ is the sequence length. This presents a significant challenge as the community moves towards LLMs that can process increasingly longer contexts. Many studies explore alternative attention mechanisms to mitigate this complexity. The Reformer architecture [31] reduces the complexity to $O(n \log n)$ and the linear attention mechanism [30, 57] further decreases complexity to $O(n)$. Recently, there has been a trend of revisiting recurrent neural networks, leading to the proposal of new architectural frameworks such as RWKV [40], RetNet [47], and Mamba [20]. Despite their promise of efficiency, these methods struggle to match the performance of full attention mechanisms, particularly with larger models and longer contexts.

**Intrinsic but Dynamic Sparsity in Attention.**    Attention mechanisms inherently exhibit sparsity, which arises from the attention map $\mathbf{A}$ generated by $\mathbf{Q}$ and $\mathbf{K}$: $\mathbf{A} = \text{softmax}(\mathbf{QK^T}/\sqrt{d})$. The softmax function often produces a multitude of negligible scores that can be treated as zeros without impacting model accuracy [59, 35, 52, 7, 36]. Attention sparsity becomes more pronounced with longer contexts, presenting opportunities to optimize inference speed. Unfortunately, this sparsity is dynamic, varying across different context inputs and attention heads, each displaying distinct sparsity locations and ratios. Prior research has attempted to approximate attention sparsity using predefined patterns and heuristics [18, 28] for different attention heads. Yet, these methods lack generality and often rely on handcrafted features, struggling to fully capture the sparsity behavior of attention mechanisms. The dynamic and input-dependent nature of attention sparsity echoes the principles of

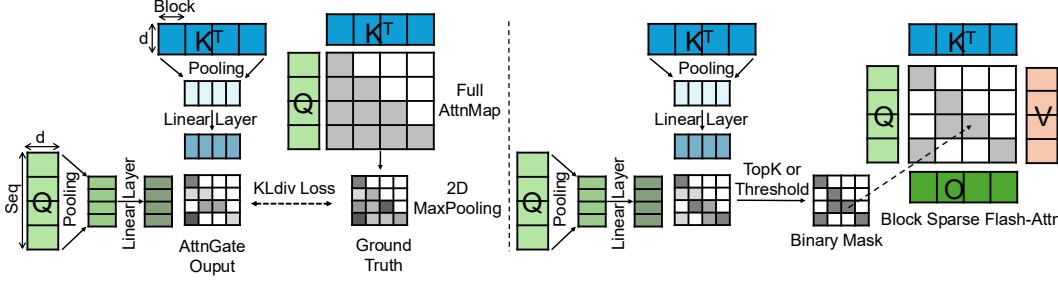

Figure 1: **Overall of SeerAttention.** The AttnGate in SeerAttention first pools the Q and K tensors in sequence dimension and passes through learnable linear layers. The AttnGate output are trained to mimic the 2D maxpooled results of the pre-trained model. During inference, it applies TopK or Thresholding on the AttnGate output to locate activate blocks.

Mixture of Experts (MoE) models [45, 17] suggesting that sparsity should ideally be learned directly from data within the model itself. This approach would allow models to adaptively harness sparsity, improving efficiency while maintaining accuracy.

**Long-Context LLM Optimizations.** The ability to process long contexts is crucial for large language models (LLMs) as it enables them to retain and utilize more extensive information. However, it comes with substantial computational and memory costs. Various research efforts have explored different strategies to optimize long-context processing. One major direction is improving prefill efficiency, where techniques such as prompt compression [27, 39, 8] and sparse attention [28, 18, 1, 62]. Another approach focuses on optimizing the decoding phase by introducing sparse loading mechanisms [56, 6]. Additionally, several methods aim to compress the KV cache, including KV cache sharing [3, 5], KV eviction policies [63, 34, 19], and KV quantization [37, 25, 14, 61].

## 3   SeerAttention

SeerAttention adopts a fully learning-based approach to adaptively identify attention sparsity in LLMs and leverages the learned sparsity for efficient inference. To ensure efficiency on modern hardware like GPUs, we focus on learning block sparsity, which can seamlessly integrate with the tiling computation scheme of FlashAttention [13, 12]. Figure 1 illustrates the overall diagram of SeerAttention, which augments conventional attention with a learnable gating module, termed *Attention Gate* (AttnGate). The AttnGate modules contain learnable parameters (linear layers) and are distilled to mimic the 2D-Maxpooled results of the attention maps. At inference time, the AttnGate can predict the block-level sparsity for the subsequent attention computation with a block-sparse FlashAttention kernel, which significantly enhances performance by reducing I/O and computation overhead.

### 3.1   Attention Gate Design

The AttnGate module is designed to learn block-wise information with minimal overhead. It takes the original matrices $\mathbf{Q}$ and $\mathbf{K}$ as inputs and downsamples them using pooling operations along the sequence dimension. As shown in Figure 1, for a given attention head, the sizes of the pooled $\mathbf{Q}$ and $\mathbf{K}$ become $[seq/B, d]$, where $B$ is the kernel and stride size of the pooling operation (non-overlapped blocks). The downsampled $\mathbf{Q}$ and $\mathbf{K}$ are then processed through a linear layer and multiplied together, similar to the standard attention operation. This results in a matrix of size $[seq/B, seq/B]$, where each element corresponds to one block in the original full attention map. With a typical block size of 64, the output of the AttnGate module is only $\frac{1}{4096}$ the size of the original attention map, making it super efficient to compute. To its simplest form, the AttnGate output soft score can be expressed as:

$$\mathbf{Q}_c = \mathrm{RoPE}\Big(W_q \ \mathrm{concat}_{i=1}^{m_q} P_i^{(q)}(\mathbf{Q}_{nope})\Big), \tag{1a}$$

$$\mathbf{K}_c = \mathrm{RoPE}\Big(W_k \ \mathrm{concat}_{j=1}^{m_k} P_j^{(k)}(\mathbf{K}_{nope})\Big), \tag{1b}$$

$$\mathbf{O} = \mathrm{softmax}(\mathbf{Q}_c\,\mathbf{K}_c^\top/\sqrt{d}). \tag{1c}$$

where $P_i^{(q)}$ and $P_j^{(k)}$ represents different pooling operations for $\mathbf{Q}$ and $\mathbf{K}$, and $d$ is the hidden size of the tensors similar to attention computation. The detailed algorithm will be explained as follows.

**Pooling Method Selection.** Pooling operations downsample tensors and may lead to information loss. To better preserve the characteristics of the attention tensors, SeerAttention allows different pooling methods to be composed for $\mathbf{Q}$ and $\mathbf{K}$. Specifically, we consider average, max, and min pooling. When applying multiple pooling methods to either $\mathbf{Q}$ or $\mathbf{K}$, the resulting pooled tensors are concatenated along the hidden dimension before being fed into the subsequent linear layer. Figure 2 presents the test perplexity on the PG19 [42] dataset for the top 15 pooling combinations using the LLaMA-3.1-8B model. We observe that applying AvgPooling on $\mathbf{Q}$ and a com-

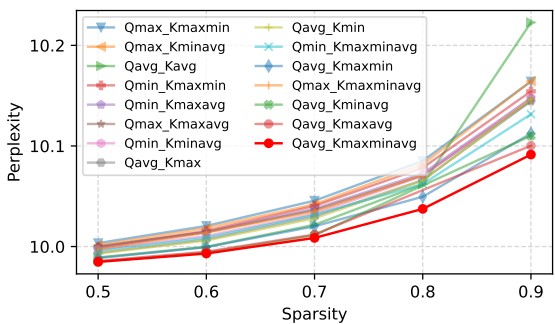

Figure 2: Test perplexity of different pooling combinations on PG19. The best configuration applies Avg-Pooling on $\mathbf{Q}$ and a combination of **Max**, **Min**, and **AvgPooling** on $\mathbf{K}$.

bination of Max, Min, and AvgPooling on $\mathbf{K}$ yields the best perplexity across different sparsity ratios. This trend may relate to prior findings in LLM quantization, where $\mathbf{K}$ tensors exhibit more outliers. Incorporating Max and Min pooling thus helps capture these extreme activations, leading to richer feature representations after pooling.

**Length Extrapolation of AttnGate using Positional Encoding.** Recent state-of-the-art LLMs typically employ positional encoding (PE) such as RoPE [46] to encode positional information. If the AttnGate relies solely on the original RoPE in the model, i.e., feeding the AttnGate with $\mathbf{Q}_{rope}$ and $\mathbf{K}_{rope}$, the positial information can possibly be damaged because of the pooling operation. This compromises the length extrapolate ability of AttnGate during distillation. To address this issue, we re-apply block-level PE in AttnGate with input wihtout PE, $\mathbf{Q}_{nope}$ and $\mathbf{K}_{nope}$, (shown in Equation 9). To represent the block-level information, the RoPE in AttnGate uses a reduced $\theta' = \theta/B$, where $\theta$ is the original RoPE theta of the LLM.

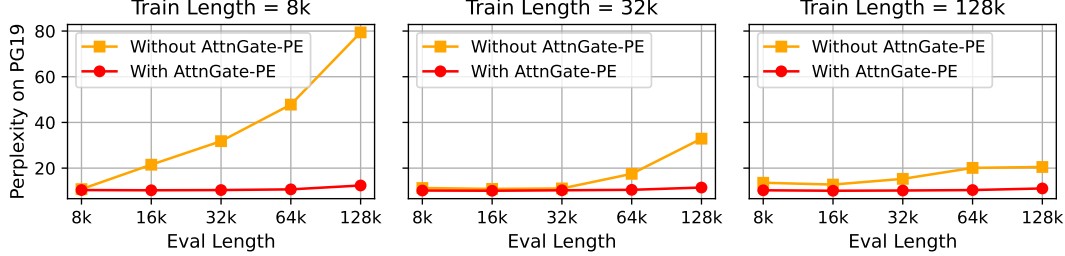

Figure 3: **Perplexity Comparison Between two PE setting in AttnGate on PG19 dataset.** The block-level RoPE in AttnGate allows it to effectively learn the block-level positional information, resulting better test performance for different context length. Without the AttnGate PE, it fails to deliver reasonable results with data longer than training length.

Figure 3 presents the test perplexity results with and without the block-level RoPE design in AttnGate. The results indicate that without this block-level RoPE design, AttnGate fails to perform adequately on evaluation data longer than 8k when trained with 8k length data. Similarly, when trained with 32k

length data, it does not perform well on 128k length data. However, with the additional block-level RoPE, AttnGate can extrapolate to different context lengths, significantly enhancing the model performance and training efficiency.

## 3.2  AttnGate Training

While the introduced SeerAttention architecture is straightforward, training presents challenges. Jointly training the gate and model from scratch, as in MoE, is costly and difficult. Fortunately, unlike MoE, where gating network must learn expert selection from scratch, the AttnGate in SeerAttention has a ground truth from standard attention for distillation.

**Obtaining the Ground Truth.**   We use the 2D-MaxPooled attention map from full attention as ground truth to distill AttnGate, as illustrated in Figure 1. Semantically, it means that only when all the attention score in a block is small, the 2D-MaxPooled results will be small, which aligned with the block-sparse definition. However, obtaining the max-pooled attention map for training is non-trivial especially in long-context scenarios due to quadratic memory consumption of the intermediate $\mathbf{Q}\mathbf{K}^T$ results. To address this challenge, we customize an efficient kernel that directly outputs the MaxPooled attention map ground truth by modifying FlashAttention kernel but largely reuses its original computation flow. The detailed design are explained in A.1.

**Loss Function.**   The Kullback-Leibler divergence loss [29] is use to distill the AttnGate. Since AttnGate uses softmax in output similar to full attention computation, the row summation of gating score will always be 1. KL-divergence loss allows the training process to focus on mimicking the attention distribution instead of absolute magnitude like Mean-square-error loss. The overall distillation process can be expressed as:

$$\mathbf{gt} = \text{MaxPool2D}\Big(\text{softmax}\big(\frac{\mathbf{Q}_{rope}\mathbf{K}_{rope}^T}{\sqrt{d}}\big)\Big), \tag{2}$$

$$\mathbf{o} = \text{AttnGate}(\mathbf{Q}_{nope}, \mathbf{K}_{nope}), \tag{3}$$

$$\mathbf{loss} = D_{KL}\big(\mathbf{gt} \parallel \mathbf{o}\big). \tag{4}$$

## 3.3  Inference with SeerAttention

After self-distillation training process, SeerAttention can utilizes the trained AttnGate to generate a gating score for each block within the full attention mechanism. These scores are then used to select the final activated sparse blocks. Further combined with our backend Block-sparse FlashAttention kernel, SeerAttention can achieve significant speedup for long-context prefilling while maintaining high accuracy.

**Generating Binary Block Mask.**   SeerAttention provides the flexibility to convert the floating-point gating scores $\mathbf{o}$ into a final binary block mask using either the TopK or Thresholding methods. If using the TopK method, the $k$ blocks with the highest scores in each row are selected.

$$b_{ij} = \begin{cases} 1 & \text{if } j \in \text{TopK}(\mathbf{o}_i, k).\text{index}, \\ 0 & \text{otherwise}. \end{cases} \tag{5}$$

Alternatively, users can activate blocks with score exceeding a threshold. This can further saving the need of sorting the AttnGate output score.

$$b = \mathbf{o} > threshold \tag{6}$$

Notably, once AttnGate is trained, for the inference stges, we can adjust the TopK ratio or threshold-based at test time to achieve various trade-offs.

**Block Sparse Flash-Attn Kernel.**   In designing the Block Sparse Flash-Attention kernel, the block size of AttnGate is aligned with the tiling size used in Flash-Attention, typically 64 or 128. By doing so, we can create a customized block-sparse Flash-Attention kernel that leverages the binary block mask generated by AttnGate to selectively skip the I/O and computation for unactivated blocks. This approach is highly efficient on modern GPUs, as it optimizes the processing of sparse data at the block level rather than dealing with fine-grained element-wise level, leading to significant performance gains.

# 4   Experiments

In this section, we evaluate both the accuracy and efficiency of SeerAttention. In our current experiments, block-size $B$ for the AttnGate and sparse kernel is fixed at 64 and AttnGate solely applies in the prefill stage.

**Models, Baselines and Tasks.**   We apply SeerAttention to the pre-trained models Llama-3.1-8B-Instruct and Llama-3.1-70B-Instruct [15], as well as Qwen2.5-7B-Instruct, Qwen2.5-14B-Instruct and Qwen2.5-32B-Instruct [55] in the experiments. We compare SeerAttention with three state-of-the-art sparse attention methods, MoA [18], MInference [28], and DuoAttention [54]. It shoule be noted that Llama-3.1-8B-Instruct is the only model that all the other methods provide official support/configuration. Thus, we only compare with them using Llama-3.1-8B-Instruct model. MoA uses an offline search scheme to apply static sparse patterns across different attention heads. In our experiment, we adopt their official implementation with "KV Sparsity" in 0.5 which means "Attention Sparsity" in 0.35. MInference dynamically generates sparse indices using heuristic methods for each head based on pre-defined sparse patterns. We used their official configuration for Llama-3.1-8B-Instruct model, where all attention heads choose the "Vertical-Slash" sparsity pattern. DuoAttention differentiates some attention heads as streaming heads [53] while keep the rest as dense heads. In the following experiment, we adopted their official setup for Llama-3.1-8B-Instruct model with 50% head as streaming heads. We evaluate the performance using on two long context benchmarks: LongBench [4] and RULER [26], and 4 short-context task from Open LLM Leaderboard [50]: HellaSwag [60], MMLU [24], ARC-challenge [9], GSM8K [10]. For long-context benchmark, we follow a similar practice in Star Attention [2] and SCBench [33] that only applies sparsity in context rather than question in SeerAttention. All the evaluation were run on A100 GPUs.

**Distillation Training Setup.**   We use the RedPajama [11] dataset for AttnGate distillation, which are chunked into 64k with BOS and EOS tokens. Our training employs a learning rate of 1e-3 with cosine decay. We set the global batch size to 16 and conduct training for only 500 steps, leveraging DeepSpeed [43] stage 2 optimization on A100 GPUs. As only AttnGate parameters are learned and updated, the distillation process can be completed within around 40 A100 hours for 7B or 8B models. To prevent the quadratic memory explosion that occurs when saving the intermediate attention map for ground truth generation, we customized a FlashAttention kernel. This kernel directly outputs the 2D max-pooled ground truth on top of the original attention computation. Further details about this kernel can be found in A.1.

## 4.1   Accuracy of Evaluation

**LongBench Evaluation.**   LongBench is a long-context understanding benchmark. We compare with those of MoA, MInference, and DuoAttention using the Llama-3.1-8B-Instruct model. DuoAttention

Table 1: LongBench Results on Llama and Qwen models.

| Model | Method | 0-4k | 4-8k | 8k+ | Avg. Acc. | Avg. Sparsity |
|---|---|---|---|---|---|---|
| | Full Attention | 55.32 | 53.98 | 52.9 | 54.07 | 0.0 |
| | MInference | 55.23 | 53.78 | 52.18 | 53.73 | 0.31 |
| Llama-3.1-8B-Instruct | MoA | 50.74 | 49.84 | 51.89 | 50.82 | 0.35 |
| | DuoAttention | 53.77 | 52.17 | 51.27 | 52.40 | 0.5* |
| | **SeerAttention** | **55.43** | **54.49** | **52.69** | **54.20** | 0.50 |
| Llama-3.1-70B-Instruct | Full Attention | 58.32 | 57.29 | 57.32 | 57.64 | 0.0 |
| | SeerAttention | 57.83 | 56.07 | 55.61 | 56.50 | 0.62 |
| Qwen2.5-7B | Full Attention | 53.72 | 50.52 | 48.21 | 50.81 | 0.0 |
| | SeerAttention | 53.94 | 50.78 | 48.73 | 51.80 | 0.55 |
| Qwen2.5-14B | Full Attention | 54.64 | 53.16 | 50.68 | 52.83 | 0.0 |
| | SeerAttention | 54.55 | 52.84 | 51.21 | 52.86 | 0.55 |
| Qwen2.5-32B | Full Attention | 56.29 | 52.17 | 51.83 | 53.43 | 0.00 |
| | SeerAttention | 56.43 | 51.93 | 52.01 | 53.45 | 0.56 |

* 50% streaming heads, the real sparsity <50%

uses 50% of the heads as streaming heads and 50% as dense heads. For streaming heads, the attention only occurs in the attention sink and recent tokens. As a result, it is not less than 50% sparsity overall. In this benchmark test, SeerAttention employs a threshold of 2e-3 for all AttnGates. With the same threshold, different attention gates can exhibit varying sparsity ratios, and longer context data tends to be sparser. This approach allows for a more adaptive allocation of sparsity. As demonstrated in Table 1, SeerAttention consistently outperforms other methods across various test lengths. Notably, in the 0-4k and 4-8k tests, our score surpasses even the dense baseline. This may be attributed to AttnGate filtering out noisy attention in certain cases. Furthermore, SeerAttention achieves the highest average score and the highest average sparsity across all tests. For other models except Llama-3.1-8B-Instruct, SeerAttention demonstrates similar accuracy performance compared to dense baseline with $> 50\%$ averaged sparsity. The 8k+ split typically has an sparsity around 70%.

Table 2: RULER Benchmark Results on Llama-3.1-8B-Instruct Model.

| Methods | 4k | 8k | 16k | 32k | 64k | 128k | Average Accuracy | Average Speedup |
|---|---|---|---|---|---|---|---|---|
| Full Attention | 95.53 | 92.37 | 92.01 | 87.63 | 84.39 | 76.26 | 88.01 | 1.00 |
| MInference | 95.53 | 92.64 | 91.37 | 85.71 | 83.24 | 67.02 | 85.92 | 0.83 |
| DuoAttention | **95.64** | 92.08 | 90.71 | 84.75 | 83.24 | **75.32** | 86.96 | 1.09 |
| **SeerAttention** | 95.53 | **92.71** | **92.02** | **88.49** | **83.48** | 73.37 | **87.60** | **1.41** |

**RULER Evaluation.** RULER is a long-context LLM evaluation benchmark consisting of 13 challenging sub-tasks. It generates tests with data sizes ranging from 4k to 128k. In this experiment, SeerAttention employs a threshold of 5e-4, which allows it to automatically adapt sparsity from approximately 10% for 4k data to around 85% for 128k data. Due to out-of-memory (OOM) issues in some tests, MoA was excluded from this benchmark. Table 2 provides detailed accuracy results across different evaluation lengths. It is evident that SeerAttention achieves the best accuracy in most tests (8k-64k). For the 128k test, DuoAttention has less than 50% sparsity, whereas SeerAttention maintains an sparsity higher than 80%, which accounts for the slightly lower performance. SeerAttention also attains the highest average accuracy compared to other models (only 0.41% lower than the dense baseline) while delivering the highest average end-to-end speedup ($1.41\times$) in prefilling time.

Table 3: Short Context Tests on Llama-3.1-8B-Instruct Model

| | MMLU | HellaS. | ARC-c | GSM-8K |
|---|---|---|---|---|
| Full Attention | 68.1 | 80.1 | 60.7 | 75.7 |
| SeerAttention | 67.9 | 79.8 | 60.2 | 75.6 |
| Avg Sparsity | 3.4 | 50.4 | 26 | 52.1 |
| Avg Seqlens | 118 | 840 | 395 | 872 |

**Short Context Test.** For short context input, attention contributes a smaller proportion in the total runtime. Consequently, sparse attention does not significantly enhance latency performance. Nevertheless, we evaluate SeerAttention accuracy performance under a very high threshold 3e-2 to achieve high sparsity. The results, as shown in 3, indicate that SeerAttention exhibits negligible accuracy loss. For instance, with an average sequence length of 872 in the GSM-8K task, SeerAttention achieves only 0.1% degradation in accuracy with 52% averaged sparsity.

## 4.2 Efficiency Evaluation

We evaluate the efficiency of SeerAttention using our implementation of CUDA kernels. We evaluate the kernel-level as well as end-to-end speedup using a Llama-3.1-8B-Instruct on a single A100 GPU. Results are compared to FlashAttention-2 (dense baseline), MoA, MInference and DuoAttention.

### 4.2.1 Kernel evaluation

**Negligible Overhead incurred by AttnGate.** 4 shows the kernel-level latency breakdown of SeerAttention. It demonstrates that the overhead introduced by the AttnGate during inference is minimal. For instance, at a context length of 32K and a sparsity of 0.5, the AttnGate contributes

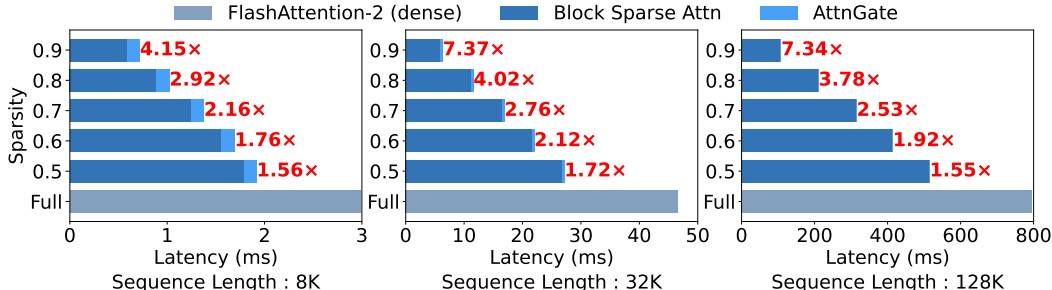

Figure 4: **SeerAttention Speedup over FlashAttention-2 at the Kernel Level.** The latency overhead from AttnGates is minimal. Our block-sparse attention kernel achieves highly linear speedup over dense configurations, delivering a 7.3× speedup with 90% sparsity on sequences of 128k. The AttnGate overhead amost diminishes in 128k context length.

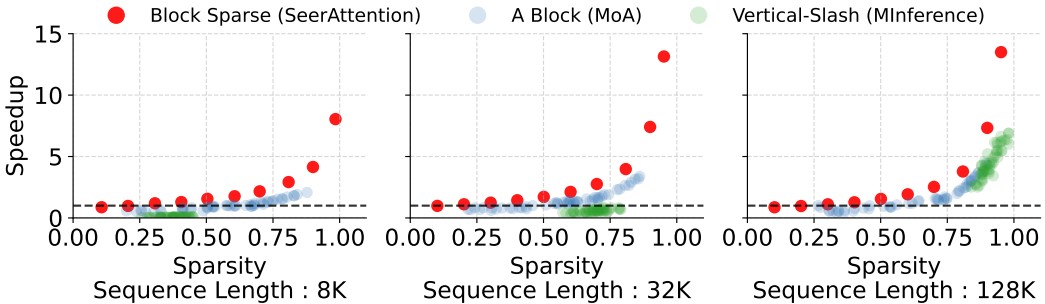

Figure 5: Kernel-level Speedup Comparison Between Different Works. SeerAttention translates sparsity to speedup more effectively.

only 1% to the total latency of an attention layer. In the cases of 128K sequence length, the relative overhead almost diminishes.

**Block-sparse FlashAttention Kernel Speedup.** Figure 4 also shows that our kernel exibits linear speedup over various sparsity levels. At a sequence length of 128K with 90% sparsity, SeerAttention achieves a speedup of 7.3× compared with FlashAttention-2 (full attention) on a single A100 GPU. This demonstrates the effectiveness of the block-level sparsity employed by SeerAttention, which is highly efficient on GPUs and translates into high speedup.

**Kernel-level Comparison with Related Works.** We compare the kernel-level speedup of SeerAttention with MoA and MInference. MInference uses offline calibration to identify a pre-defined sparse pattern for each layer. For Llama-3.1-8B-Instruct model, MInference consistently uses "Vertical-slash" pattern across all layers. During runtime, MInference will dynamically generate non-zero indices based on their approximation algorithm. On the other hand, MoA uses "A-shape" blocks as their sparse pattern and calibrate the shape parameters offline under given sparsity constraint. DuoAttention is omitted in kernel-level comparison as it's a combination between streaming and dense head, whose performance is a mixture results of block sparse attention and dense FlashAttention.

Figure 5 shows the sparsity v.s. speedup plots of different methods on 8k, 32k, 128k sequences length, where the speedup baseline is FlashAttention-2. The kernel-level sparsity statistics were collected from PG19 datasets. For MoA, we generated the sparse configurations under their 0.5 overall "KV-sparsity" constraints, which corresponds to an average of 0.35 sparsity in attention. The results demonstrates that the block-sparse attention kernel used in SeerAttention outperforms both MoA and MInference in most cases.

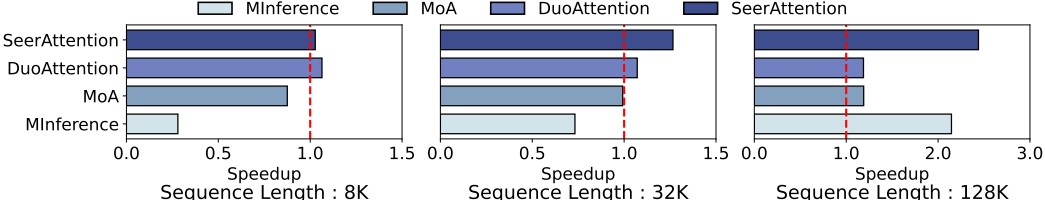

Figure 6: **Comparing Prefilling Time Speedup on RULER Test Setting.** SeerAttention outperforms related works in most long-context data scenarios ($\geq$ 16k). For longer context data, the attention mechanism constitutes a larger proportion of the total runtime, allowing sparse methods to achieve better speedup. Overall, SeerAttention achieves the highest average speedup ($1.41\times$) while maintaining the best average accuracy under this RULER benchmark setting.

#### 4.2.2 End-to-end Speedup Comparison.

To assess the end-to-end speedup of our method, we measured the average prefilling time, or time-to-first-token (TTFT), using the Llama-3.1-8B-Instruct model on the RULER test discussed above. Since attention takes up more runtime with longer contexts, all methods generally achieve better speedup with longer context lengths. It should be noted that SeerAttention uses an identical threshold across all tests in RULER, automatically adjusting to higher sparsity for longer contexts (ranging from approximately 10% sparsity for 4k to around 85% sparsity for 128k). This approach results in an end-to-end prefilling speedup of up to $2.43\times$ on 128k length. On the other hand, MInference experiences a slowdown with data sizes less than 64k due to significant overhead in searching for sparse indices during runtime. It is feasible for SeerAttention to adjust to higher sparsity to achieve greater speedup in shorter contexts as a tradeoff. Nevertheless, SeerAttention delivered the highest average accuracy (87.6) and the greatest average speedup ($1.41\times$) in this RULER benchmark setting.

### 4.3 Training Cost and Parameter Overhead.

The additional training cost of SeerAttention is modest. On LLaMA-3-8B-Instruct, it requires about 40 A100 GPU hours, similar to or lower than DuoAttention. In comparison, MInference calibrates faster but shows lower accuracy and slower inference. SeerAttention introduces about 101 M trainable parameters, roughly 1.3 % of the model, and adds less than 5 % memory and latency overhead compared with FlashAttention, supported by our customized distillation kernels. The gating module also scales well to larger models, such as 503 MB for LLaMA-3.1-70B and 252 MB for DeepSeek-R1-Distill-Qwen-32B, showing that the method remains lightweight and scalable for large deployments.

### 4.4 Visualization of Learned Attention Maps.

The AttnGate module automatically learns diverse sparse patterns without any prior knowledge or heuristics. Figure 7 shows several example outputs from AttnGate, including (a) "A-shape," or streaming head (b) "Vertical," (c) "Slash" with empty vertical spaces, (d) block sparsity along the diagonal, and (e) random patterns. These patterns not only encompass but also extend beyond those observed in previous works such as MoA and MInference, showcasing the versatility of our learning based methods.

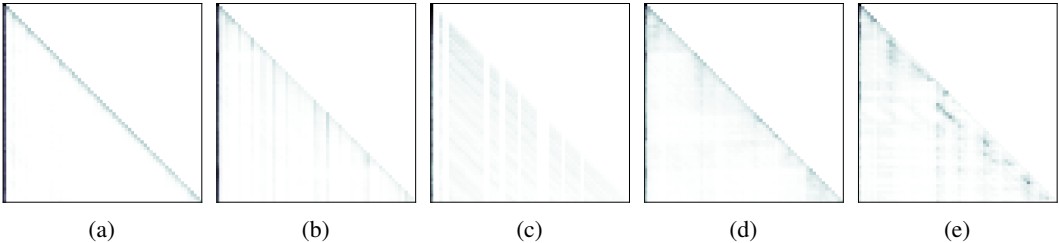

Figure 7: Visualization of the AttnGate's outputs.

## 5 Conclusion and Future Work

This paper presents SeerAttention, a new attention mechanism that learns and leverages the intrinsic sparsity in attention to boost long-context LLMs. SeerAttention learns the attention sparsity from the LLM itself with a lightweight self-distillation approach. Our experiments demonstrate that SeerAttention outperforms previous approaches in terms of long context model accuracy and pre-filling latency. For future work, there are several promising directions to explore for improving and expanding the capabilities of SeerAttention. One key area is enhancing the training methodologies for SeerAttention, such as applying SeerAttention in long-context continued pre-training with more training tokens to achieve higher sparsity without sacrificing accuracy. Another important avenue is applying SeerAttention in the decoding stage, especially for long-CoT.

## Acknowledgments

We gratefully acknowledge Prof. Ang Li from University of Washington for his help with the paper revision and valuable feedback.

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

# A  Appendix

## A.1  Training SeerAttention with Customized GPU Kernel

In this appendix, we provide a detailed design and implementation of our efficient kernel, highlighting key modifications to FlashAttention and optimizations for long-context scenarios. We then evaluate the peak memory usage and additional latency overhead of our training kernel during the AttnGate training stage, showing that it incurs only minimal overhead in both memory and latency compared to training with FlashAttention-2.

**FlashAttenion with 2D-MaxPooling: A Customized training kernel.**  In Section 3.2, we discussed the method for obtaining the ground truth attention map used to distill AttnGate. Specifically, we leverage the 2D-MaxPooled attention map from full attention as the ground truth, aligning with the block-sparse attention definition. However, directly computing this attention map is challenging due to the quadratic memory complexity and the fused operation nature of FlashAttention. To overcome this, we developed a customized kernel based on Triton [49] that efficiently extracts the 2D-MaxPooled attention map by modifying the FlashAttention kernel while largely preserving its computation flow. Figure 8 shows the pseudo code and diagram of this customized kernel.

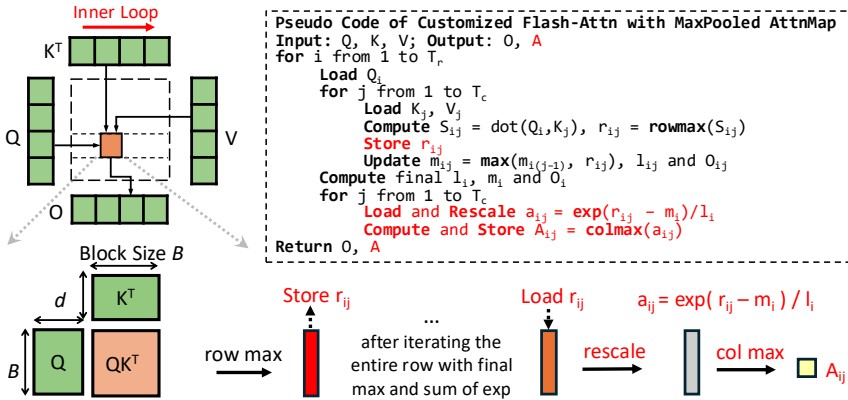

Figure 8: Efficient FlashAttention kernel with pooling of attention map.

Normally, the softmax function ensures numerical stability by subtracting the maximum value before applying the exponential operation. FlashAttention computes the local row max of each block, and gradually updates the global maximum through iteration:

$$S_{ij} = Q_i K_j^T;$$
$$r_{ij} = \text{rowmax}(S_{ij}); \qquad\qquad (7)$$
$$m_{ij} = \max(m_{i(j-1)}, r_{ij}).$$

where $r_{ij}$ is typically treated as a temporary result. However, we store it in HBM and rescale it later with the final global max $m_i$ and sum of exp $l_i$ after the iteration:

$$a_{ij} = \exp(r_{ij} - m_i)/l_i \qquad\qquad (8)$$

This $a_{ij}$ represents the correct row max of the original attention block. With that, 2D-MaxPooling is achieved by applying a column max over $a_{ij}$. This introduces only minor overhead (storing and rescaling $r_{ij}$) but significantly improves the efficiency of obtaining the ground truth. The overhead of memory and latnecy analysis is in Figure 9.

**Performance of the Training Kernel.**  We evaluate our customized FlashAttention kernel with 2D-MaxPooled attention map for scalable training of SeerAttention by comparing against with PyTorch naïve manual attention implementation and FlashAttention-2. As shown in Figure 9b, the PyTorch kernel runs out of memory (OOM) when the sequence length exceeds 4k, while our customized kernel costs similar peak memory usage compared to FlashAttention-2. Regarding latency, since PyTorch encounters OOM for sequences longer than 8K, the attention operations per head into a loop to assess

kernel-level latency. Figure 9b shows that the latency overhead introduced by the additional pooling operation is minimal compared to FlashAttention-2, while the PyTorch implementation suffers from a significant slowdown.

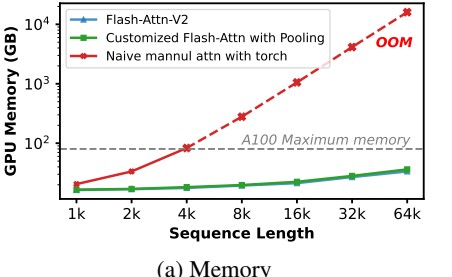

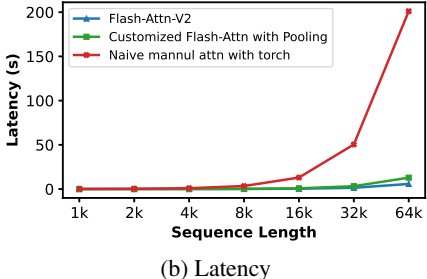

(a) Memory

(b) Latency

Figure 9: Memory and latency of customized FlashAttention with max-pooling training kernel.

## A.2 Preliminary Experiments of Fine-tuning with SeerAttention

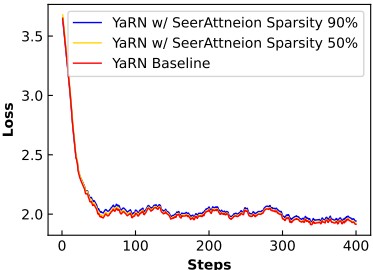

Figure 10: Fine-tuning Loss.

Figure 11: By incorporating SeerAttention with YaRN [41] to extend a Llama-3-8B model from 8k to 32k context length, the loss curves for 50% to 90% sparsity are nearly identical to the dense YaRN baseline.

Table 4: Perplexity of YaRN baseline, SeerAttention after YaRN and YaRN fine-tuning with SeerAttention.

|  | YaRN | Post-training SeerAttention after YaRN | | | | | YaRN with SeerAttention | | | | |
|---|---|---|---|---|---|---|---|---|---|---|---|
| Sparsity | 0.0 | 0.5 | 0.6 | 0.7 | 0.8 | 0.9 | 0.5 | 0.6 | 0.7 | 0.8 | 0.9 |
| PG19 | 8.79 | 9.16 | 9.30 | 9.48 | 9.73 | 10.18 | 8.81 | 8.82 | 8.85 | 8.93 | 9.16 |
| Proof-pile | 2.46 | 2.53 | 2.57 | 2.61 | 2.68 | 2.85 | 2.47 | 2.47 | 2.48 | 2.51 | 2.60 |

In this preliminary experiment, we demonstrate that SeerAttention can be seamlessly integrated in Long-context extension fine-tuning stages. We follow YaRN [41] to extend the context size of a Llama-3-8B model from 8k to 32k. The loss function is the summation of original LM loss and AttnGate loss. To ensure stable training, the AttnGates are first initialized using the post-training self-distillation before fine-tuning the entire model. We integrate SeerAttention into YaRN and compare the performance against the YaRN dense baseline and the post-training time self-distillation of SeerAttention applied after YaRN. Figure 10 presents the loss curves of the YaRN dense baseline and SeerAttention at 50% and 90% sparsity. The curve at 50% sparsity nearly overlaps with the baseline, while the curve at 90% sparsity shows slightly higher loss. Table 4 displays the test perplexity on the PG19 and ProofPile datasets evaluated at a 32k context length. The YaRN dense baseline achieves perplexity scores of 8.79 and 2.46, respectively. Post-training SeerAttention results in increased perplexity. When applying SeerAttention during the YaRN extension fine-tuning, it maintains near-lossless performance at 50% sparsity (with scores of 8.81 and 2.47), and even at 90% sparsity, the loss remains minimal.

### A.3 Preliminary Results on Introducing Sparse Attention at Decoding Stage

**Adjusting AttnGate for decoding**  The current AttnGate design mainly works for accelerating long-context prefill. However, applying the attention gate distillation idea is also a feasible direction. This is important to improve the efficiency of reasoning models that generate longer sequences during inference before producing an answer, aka test-time scaling. Adjusting current design to compatible for decoding cases requires removing the sequence level pooling of Query to adhere to the token-by-token generation fashion. Here is a modification of AttnGate design in (9):

$$\mathbf{Q}_c = \text{RoPE}\Big(\mathbf{W}^{\mathbf{q}}_{\mathbf{gate}} \ \text{reshape}(\mathbf{Q}_{nope}, [..., g \cdot d])\Big), \tag{9a}$$

$$\mathbf{K}_c = \text{RoPE}\Big(\mathbf{W}^{\mathbf{k}}_{\mathbf{gate}} \ \text{concat}[\text{P}_{\max}(\mathbf{K}_{nope}), \text{P}_{\min}(\mathbf{K}_{nope}), \text{P}_{\text{avg}}(\mathbf{K}_{nope})]\Big), \tag{9b}$$

$$\mathbf{S} = \text{softmax}(\mathbf{Q}_c \, \mathbf{K}_c^{\top} / \sqrt{d_{gate}}). \tag{9c}$$

where, $\text{P}_{\max}$, $\text{P}_{\min}$, and $\text{P}_{\text{avg}}$ stand for Max, Min and Average Pooling in sequence dimension, and $g$ is the group size of GQA setting. $d$ and $d_{gate}$ are the hidden dimension of the original model and AttnGate for each head, respectively. $\mathbf{S}$ is the output score of each block from AttnGate. We aggregate Query heads within each group to share sparsity decisions to improve the decoding efficiency. Specifically, a linear layer in the $\mathbf{Q}$ branch reduces each subgroup of queries (e.g., 32 heads $\rightarrow$ 8 heads for $g$=4) to a single $\mathbf{Q}_c$ head while keeping $\mathbf{K}$ heads unchanged, enabling shared sparsity among grouped queries. To compress $\mathbf{K}$ along the sequence dimension, we apply non-overlapping block-level pooling that concatenates Max, Min, and Average pooling outputs before a linear projection. Additionally, AttnGate reapplies RoPE on pre-RoPE $\mathbf{Q}$ and $\mathbf{K}$, assigning each block the position of its first token, which we find improves the accuracy.

**Evaluation on Reasoning Tasks**  We evaluate this design on three reasoning benchmarks: AIME24, MATH-500 [23], and GPQA-Diamond [44] using DeepSeek-R1-Distill-Qwen-14B [21]. We compare against full attention and Quest [48], which is a training-free sparse decoding method using query-aware KV cache selection. Both methods use a block size of 64 and apply sparsity to all layers for fair comparison. The maximum output length is fixed at 32,768 tokens across all settings. Accuracy is reported as average pass@1 over 64 (AIME24), 8 (MATH-500), and 16 (GPQA) samples. For AttnGate distillation, we use OpenR1-MATH-220k [16] dataset with 800 steps and global batch size 16 on AMD MI300x GPUs, employing DeepSpeed ZeRO-2, AdamW optimizer, and a 1e-3 learning rate with cosine decay.

Table 5: Performance comparison across different token budgets for SeerAttention-decoding and Quest using DeepSeek-R1-Distill-Qwen-14B model.

| Method | Dataset | Token Budgets | | | | Full |
|---|---|---|---|---|---|---|
| | | **2k** | **4k** | **6k** | **8k** | |
| SeerAttention-decoding | AIME24 | 55.78 | 66.35 | 67.50 | 66.82 | 67.50 |
| | MATH500 | 87.65 | 92.10 | 93.05 | 93.12 | 93.30 |
| | GPQA-Diamond | 51.26 | 56.79 | 56.41 | 57.48 | 57.80 |
| Quest | AIME24 | 25.83 | 46.67 | 53.75 | 60.00 | 67.50 |
| | MATH500 | 57.40 | 79.60 | 88.60 | 92.20 | 93.30 |
| | GPQA-Diamond | 35.35 | 50.25 | 52.53 | 54.55 | 57.80 |

Table Table 5 shows the superior performance of our design compared to Quest across multiple reasoning benchmarks and token budgets. On AIME24, we consistently outperforms Quest, achieving 55.78 vs. 25.83 accuracy at 2k tokens and maintaining a strong lead using more token budgets. Similarly, on MATH500 and GPQA-Diamond, SeerAttention-decoding shows better budget-accuracy tradeoff compared with Quest, reflecting better information retention under sparse decodin owing to its learned gate that shares sparsity decisions, enabling more coherent block selection.

