# OpenReview forum: "SeerAttention: Self-distilled Attention Gating for Efficient Long-context Prefilling"
_NeurIPS.cc/2025/Conference — NeurIPS 2025 poster_

### Official Review · Reviewer_HW7W · 2025-06-09

**Clarity:** 3
**Significance:** 2
**Originality:** 2
**Rating:** 4
**Confidence:** 4

**Summary:**

This paper introduces SeerAttention, which employs a learnable method to achieve sparsity of attention computations. The paper also introduces self-distillation and kernels to achieve efficient training and inference. Experiments show that the method can achieve better efficiency while preserving the performance.

**Questions:**

Seeing Weaknesses

**Ethical Concerns:**

["NO or VERY MINOR ethics concerns only"]

**Final Justification:**

The authors respond to most of my questions and weaknesses.

**Limitations:**

Yes

**Quality:**

3

**Strengths And Weaknesses:**

Strengths:
* This paper introduces a new sparse attention, which can be obtained through distillation from the full attention.
* The authors design kernels to achieve efficiency.
* Experiments show the performance and efficiency of this method.

Weaknesses:
* **Lack of differences from other methods**: from the perspective of novelty, the core idea is similar to NSA and MoBA. In addition, I believe these attention variants can also be integrated into existing models via self-distillation rather than training them from scratch, as the authors proposed.
* **Unfair comparison**: Compared with baselines, the trainable parameters are larger, and knowledge distillation is also employed in SeerAttention. Thus, I think the comparisons are not fair. I suggest that the authors can train the models with baselines using similar distillation methods.
* **Lack of ablation studies**:
    * w/o self-distillation
    * Different settings of sparsity and block size.

---

> ### Author Rebuttal · Authors · 2025-07-29
>
> We sincerely thank the reviewer for the constructive feedback and valuable suggestions. We address the main concerns below:
>
> ### **1. Lack of differences from other method**
>
> We respectfully disagree with the assessment that our method lacks novelty. SeerAttention introduces learnable sparse attention specifically for post-training, requiring only a small number of additional gate parameters that are distilled during this stage. In contrast, NSA and MoBA adopt a similar idea but focus on the pre-training phase. As shown in the comparison of trainable parameters and training tokens, NSA and MoBA require training all model weights with a large number of tokens and extensive GPU hours. Our approach, by contrast, is a lightweight and cost-effective self-distillation method.
>
> | Base Model                   | SeerAttention (trainable parameters)  |Training Tokens|
> | ---------------------------- | ------------- |------------- |
> | LLaMA‑3.1‑8B‑Instruct| 101 MB  | 0.5B|
> | LLaMA‑3.1‑70B‑Instruct| 503 MB        |0.5B|
> | Qwen2.5‑7B‑Instruct| 77 MB         |0.5B|
> | Qwen2.5‑14B‑Instruct| 189 MB|0.5B|
> | Qwen2.5‑32B‑Instruct| 252 MB        |0.5B|
> | DeepSeek‑R1‑Distill‑Qwen‑7B  | 101 MB|0.5B|
> | DeepSeek‑R1‑Distill‑Qwen‑14B | 189 MB        |0.5B|
> | DeepSeek‑R1‑Distill‑Qwen‑32B | 252 MB        |0.5B|
>
> | Base Model                   | MoBA (trainable parameters)  |Training Tokens|
> | ---------------------------- | ------------- |------------- |
> |  568M | 568M (whole)  | 10.8B |
> |  822M | 822M (whole)  |  15.3B |
> |  1.5B | 1.5B (whole)  |  27.4B |
> |  Llama-8B-1M| 16 GB (whole)  | 36.9B|
>
> | Base Model                   | NSA (trainable parameters)  |Training Tokens|
> | ---------------------------- | ------------- |------------- |
> |  7B-parameter transformer  | 7B (whole)  | 260B |
>
> We have discussed this point with another reviewer, as seen in this comment (https://openreview.net/forum?id=Nf8yfPDFTl&noteId=LV1xhwZYfg). In fact, both of the two works have recognized acknowledged the novelty of SeerAttention. (ACs could help verify this if needed.) Therefore, we believe our work is indeed novel and offers a new direction for developing sparse attention methods.
>
> ### **2. Fairness of comparisons (trainable parameters and self-distillation)**
>
> We would like to clarify that the “self‑distillation” in our method is different from conventional knowledge distillation as understood in the review. Our self‑distillation **only trains the newly introduced gate parameters, while keeping the original model weights entirely frozen**. The additional parameters are **minimal**—for example, in LLaMA‑3.1‑8B, the gate introduces approximately **101M** parameters (about **101M/8B = 1.3%** of the original model size); see also our response to Reviewer DFJm (https://openreview.net/forum?id=Nf8yfPDFTl&noteId=yPnYUXTMey).
>
> As illustrated in Figure 1, this distillation process simply guides the gate to mimic the attention pattern of the frozen full‑attention model, here we call it "self‑distillation". Hence, no additional knowledge distillation is applied to the backbone model itself. For this reason, it is not meaningful to ***“train the baselines with similar distillation methods,”*** since our approach does not perform distillation on the original model’s weights.
>
> ### **3. Different settings of sparsity and block size.**
>
> We appreciate the suggestion to include experiments with different sparsity and block size settings.
>
> For **sparsity**, SeerAttention allows runtime adjustment via the threshold T, providing flexibility to trade off efficiency and accuracy. The table below shows that as T increases, sparsity rises while accuracy decreases slightly, and SeerAttention consistently outperforms DuoAttention at comparable sparsity.
>
> | Method                           | Sparsity | Accuracy (%) |
> | -------------------------------- | -------- | ------------ |
> | DuoAttention 0.5                 | 0.50     | 75.32        |
> | DuoAttention 0.7                 | 0.70     | 70.35        |
> | DuoAttention 0.75                | 0.75     | 55.48        |
> | SeerAttention $T=5\times10^{-5}$ | 0.57     | 75.71        |
> | SeerAttention $T=1\times10^{-4}$ | 0.62     | 75.15        |
> | SeerAttention $T=2\times10^{-4}$ | 0.72     | 74.94        |
> | SeerAttention $T=4\times10^{-4}$ | 0.80     | 73.81        |
> | SeerAttention $T=5\times10^{-4}$ | 0.83     | 73.37        |
>
> #### **LongBench performance & efficiency results on Llama-3.1-8B-Instruct**
>
> | Methods| 0-4k  | 4-8k  | 8k+   | Avg. Accuracy | Avg. Sparsity | Avg. Speedup |
> |----------|-------|-------|-------|-----------|---------------|---------|
> |  Baseline (Full Attention)  | 55.32 | 53.98 | 52.90 | 54.07| 0.00  | 1.00x    |
> | MInference| 55.23 | 53.78 | 52.18 | 53.73     | 0.31     | 0.72x|
> | MoA| 50.74 | 49.84 | 51.89 | 50.82     | 0.35| 1.11x|
> | DuoAttention    | 53.77 | 52.17 | 51.27 | 52.40     | 0.50*   | 1.07x|
> | SeerAttention | 55.43 | 54.49 | 52.69 | 54.20 | 0.50 | 1.32x |
>
> \* 50% streaming heads, the real sparsity < 50%
>
> In RULER, we employ a threshold of 5e-4 under 4k-128k test benches.
>
> #### **RULER performance & efficiency results on Llama-3.1-8B-Instruct**
>
> | Methods       | 4k    | 8k    | 16k   | 32k   | 64k   | 128k  | Avg. Accuracy | Avg. Sparsity | Avg. Speedup |
> |---------|-------|-------|-------|-------|-------|-------|--------|---------|-------------|
> |  Baseline (Full Attention)| 95.53 | 92.37 | 92.01 | 87.63 | 84.39 | 76.26 | 88.01    | 0.00             | 1.00x            |
> | MInference    | 95.53 | 92.64 | 91.37 | 85.71 | 83.24 | 67.02 | 85.92    | 0.36 | 0.83x    |
> | DuoAttention  | 95.64 | 92.08 | 90.71 | 84.75 | 83.24 | 75.32 | 86.96| 0.52    | 1.09x|
> | SeerAttention | 95.53 | 92.71 | 92.02 | 88.49 | 83.48 | 73.37 | 87.60| 0.55| 1.41x|
>
> For **block size**, we appreciate the reviewer’s interest in this factor, which was one of the key considerations in our system and algorithm design. Intuitively, using a larger block size **reduces memory allocation overhead** because the block‑wise attention map has shape $[bs, nh, seq/block, seq/block]$. However, at the algorithmic level, a larger block size leads to more **coarse‑grained distillation**. We have already conducted ablation studies in this direction. As shown below, larger block sizes generally improve perplexity at longer sequences, with **block size = 64** providing the best overall trade‑off. Nevertheless, the optimal choice may depend on the downstream task, and we will explore additional experiments in the revision if this factor is deemed critical.
>
> | Block Size | 8k    | 16k   | 32k   | 64k   | 128k  |
> | ---------- | ----- | ----- | ----- | ----- | ----- |
> | 32*32         | 10.49 | 10.36 | 10.46 | 11.09 | 11.91 |
> | 64*64         | 10.18 | 10.01 | 10.10 | 10.29 | 10.71 |
> | 128*128        | 10.25 | 10.04 | 10.12 | 10.31 | 10.61 |
>
> We hope our responses have addressed your concerns and clarified the contributions of our work. Please let us know if there are further questions or suggestions, as we are happy to provide additional information.

---

> > ### Comment · Reviewer_HW7W · 2025-08-05
> >
> > Thank you for your responses! I will increase my rating!

---

> > > ### Author Response · Authors · 2025-08-05
> > >
> > > Thank you very much for your response! We really appreciate your support and are glad to hear that you'll be increasing your rating. We will make sure to incorporate the corresponding revision from the rebuttal into the final version of the paper.

---

### Official Review · Reviewer_ebQ8 · 2025-06-23

**Clarity:** 2
**Significance:** 3
**Originality:** 2
**Rating:** 4
**Confidence:** 4

**Summary:**

The paper proposes a dynamic block sparse attention approach to accelerate the prefilling stage of large language models (LLMs). Specifically, it pools query (Q) and key (K) matrices into blocks along the sequence dimension, projecting them via learnable linear layers, and computes inner products to predict blocks with high attention values. Only the blocks estimated to have high attention values are computed explicitly. The authors also propose a corresponding training scheme using KL-divergence to align dense and sparse attention masks.

**Questions:**

1. Could you provide additional results comparing sparse prefill with dense decoding across MoA, DuoAttention, and SeerAttention to ensure fairness and isolate the effect of sparse prefill?
2. Could you include a comprehensive figure or table illustrating the performance-efficiency trade-offs (e.g., benchmark scores versus runtime) of various sparse attention methods **across different density levels**? This would effectively demonstrate SeerAttention’s relative strengths across conditions.

**Ethical Concerns:**

["NO or VERY MINOR ethics concerns only"]

**Final Justification:**

SeerAttention delivers speed and accuracy gains under prefill-only and decoding-only sparse attention setups. Although a fully unified comparison remains lacking, I see positive signs towards this end during the rebuttal phase, thus raising my score to borderline accept. If the paper gets accepted, I strongly encourage the authors to complete the promised fair-comparison study and detail the novelty discussion in the final version.

**Limitations:**

Yes. The authors transparently acknowledge the limitation of exclusively optimizing prefilling rather than decoding. This focus is acceptable, given future work could explore efficient decoding strategies.

**Quality:**

2

**Strengths And Weaknesses:**

**Strengths**

1. **Effective design:** The method is straightforward and effective. Experimental results demonstrate that SeerAttention outperforms state-of-the-art sparse attention methods across the LongBench and RULER benchmarks.
2. **Practical speedup:** The proposed custom kernels successfully translate theoretical improvements into practical wall-clock speedups. At 80% sparsity, SeerAttention achieves a 1.41x prefilling speedup across sequence lengths of 4K-128K in the RULER benchmark, showing superior scalability with increasing sparsity.

**Weakness**

1. **Fairness of comparisons:** SeerAttention exclusively uses sparse attention in the prefilling phase, whereas baseline methods (MoA and DuoAttention) apply sparse attention during both prefill and decoding. To ensure a fair comparison, the authors should either:
   - Evaluate all methods with sparse prefill and dense decoding, isolating the impact of sparse decoding.
   - Alternatively, discuss the efficiency-performance trade-offs more comprehensively across varying input-output lengths to highlight differences due to sparse decoding.

2. **Clarity of illustrations and explanations:** The presentation can be improved for clarity:
   - It is unclear whether the parameters $W_q$ and $W_k$ in Equation 1 are newly introduced learnable parameters distinct from the original query/key projection layers. Also it probably is, the author is advised to explicitly clarify that in the paper.
   - Lines 80-81 conflate the concepts of "dynamic" (varying by input) and "heterogeneous" (varying across heads and layers).
   - The specific method used for generating attention masks during evaluation (top-k or threshold-based) is not explicitly stated.
3. **Novelty relative to prior work:** The relationship to previous works such as MInference, which also uses block-sparse attention with pooling-based block estimation, should be clarified in greater detail. Clearly articulating the differences from existing methods would strengthen the paper’s novelty claims.

---

> ### Author Rebuttal · Authors · 2025-07-29
>
> We sincerely thank the reviewer for your thoughtful and constructive feedback. We are glad that the reviewer recognized the **effectiveness, practical speedup, and scalability** of SeerAttention. The comment have helped us identify ways to improve the clarity and fairness of our presentation, and we have carefully addressed each point below.
>
> ### **1. Comparisons setting**
> Thank you for pointing out this issue. In our experiments, we directly used the original open-source implementations of MoA and DuoAttention. We agree that a more controlled comparison (using dense decoding for all methods) would be fairer. We are re-implement MoA and DuoAttention frameworks and doing the following the experiments settings: variable input length (prefilling) + variable output length (decoding) to fairly compare. Due to the rebuttal time limit, we could not re-implement and rerun all experiments with modified decoding settings.
>
>
> ### **2. SeerAttention Decoding**
>
> We appreciate the suggestion to discuss decode‑stage evaluation. In this work, we primarily focus on accelerating the prefill stage and leave decode‑stage optimization for future exploration. Prefill and decode are two distinct phases, and several PD‑disaggregation serving systems, such as DistServe [1] and Mooncake [2], have explored separating prefill and decode to improve TTFT/TPOT. Many prior works also focus on only one phase: for instance, MInference mainly targets prefill, while Quest [3] and H2O [4] focus solely on decode.
>
> Our approach can naturally extend to decode sparsity with a minor modification of the AttnGate design. Specifically, we remove the Q‑compression branch (pooling + linear) and instead distill using a 1D max‑pooling ground truth of the attention map (along the sequence dimension) from the base model. In other words, during distillation for decode, we directly use the original Q without sequence‑level compression, enabling each token’s Q to select important KV blocks at inference.
>
> We evaluated this design on the DeepSeek‑R1‑Distill‑Qwen‑14B reasoning model using two challenging math tasks and compared it against Quest. Below we provide a brief summary of the results (block size = 64). A more complete version can be included in the appendix upon acceptance. The results show that SeerAttention achieves higher accuracy than Quest, even with larger KV blocks and sparser attention.
>
> #### **Model: DeepSeek-R1-Distill-Qwen14B**
>
> | Method (**AIME**)                                       | Acc   | Avg Sparsity | Avg Decode Length |
> |----------------------------|-------|--------------|-------------------|
> | Full Attention                                   | 70    | 0            | 10k               |
> | Quest (Block Size 16, Token Budget 8192)         | 53.33 | 36%          | 17k               |
> | Quest (Block Size 64, Token Budget 8192)         | 16.67 | 65%          | 28k               |
> | SeerAttention (Block Size 64, Threshold=5e-3)    | 66.7  | 86%          | 11k               |
>
>
> | Method (**MATH500**)                                          | Acc   | Avg Sparsity | Avg Decode Length |
> |--------------------------------------------------|-------|--------------|-------------------|
> | Full Attention                                   | 94    | 0            | 4k                |
> | Quest (Block Size 16, Token Budget 2048)         | 83.4  | 73%          | 6k|
> | SeerAttention (Block Size 64, Threshold=7e-3)    | 94    | 85%          | 4k|
> | SeerAttention (Block Size 64, Threshold=1e-2)    | 92.8  | 89%          | 4k|
>
>
> ### **3. Clarity of illustrations and explanations**
> #### **3.1** Thank you for the suggestion. We will update the notation to use different superscripts to avoid confusion with the original model weights. The gate weights, $W_q^{\text{gate}}$ and $W_k^{\text{gate}}$, are newly introduced learnable parameters, and only the gate is trained during self-distillation. For LLaMA‑3.1‑8B, the gate contains 101M parameters (~1.3% of the total model parameters).
>
>
> #### **3.2** We appreciate the comment on the wording (dynamic vs. heterogeneous) and will revise it accordingly.
>
>
> #### **3.3** For mask generation, we illustrate the process using an example within a single head of one layer. Suppose the sequence length is $S$ and the block size is $B$. The output of AttnGate in this head is $o = [S/B, S/B]$.
>
>
> - For the top‑$k$ method, we apply a torch.topk operation on each row, obtaining indices with shape $[S/B, k]$.
>
>
> - For the threshold‑based method, we compute a binary mask as $mask = o > \text{threshold}$. Blocks marked as True in this mask are directly activated in our kernel.
>
>
> ### **4. Novelty relative to prior work**
>
> Block-sparse attention indeed dates back to much earlier works such as BigBird [5] and BlockSparse-BERT [6], and MInference is also a more recent approach that adopts block-sparse attention with pooling-based block estimation. However, our method is fundamentally different: pooling-based selection itself is not the core contribution here. The key novelty lies in introducing **a learnable gating mechanism for sparse attention**, similar to **how a trainable router in MoE outperforms static pruning or heuristic compression methods**. Our approach enables dynamic and data-dependent sparsity decisions guided by full-attention supervision, which sets it apart from heuristic or static approaches like MInference. Also, there are some following works like NSA [7] and MoBA [8] adopt this approach to pre-training stage.
>
>
> ### **5. Question 2: performance-efficiency trade-offs**
> Thank you for this helpful suggestions. We agree that our presentation of the performance–efficiency trade-off could be improved. While the paper already contains the necessary data (e.g., accuracy in **Table 2** and speedup in **Figure 6**), we did not present them together in a unified figure or table, which may have reduced clarity.
>
> To better illustrate this trade-off, we provide additional results on the 128k RULER benchmark below, showing how accuracy varies with sparsity under different thresholds (T = threshold):
>
> | Method| Sparsity | Accuracy (%) |
> | ---------------- | -------- | ------------ |
> | DuoAttention 0.5| 0.50     | 75.32 |
> | DuoAttention 0.7| 0.70     | 70.35|
> | DuoAttention 0.75| 0.75     | 55.48|
> | SeerAttention $T=5\times10^{-5}$ | 0.57  | 75.71  |
> | SeerAttention $T=1\times10^{-4}$ | 0.62| 75.15 |
> | SeerAttention $T=2\times10^{-4}$ | 0.72| 74.94|
> | SeerAttention $T=4\times10^{-4}$ | 0.80| 73.81  |
> | SeerAttention $T=5\times10^{-4}$ | 0.83| 73.37  |
>
> For the kernel-level efficiency, we measured different sequence length latency under different sparsity setting:
>
> | Sparsity / SeqLen | 8k   | 16k  | 32k  | 64k  | 128k |
> |-------------------|------|------|------|------|-------|
> | 50%               | 1.73x| 1.78x| 1.81x| 1.84x| 1.84x |
> | 70%               | 2.58x| 2.80x| 2.94x| 3.01x| 3.01x |
> | 90%               | 5.91x| 6.93x| 7.87x| 8.28x| 8.63x |
>
> For tradeoffs between speedup and accuracy, we employ a threshold of 5e-4 for all AttnGates under 4k-128k test benches and count the average sparsity and accuracy to present RULER performance & efficiency results on Llama-3.1-8B-Instruct.
>
> | Methods       | 4k    | 8k    | 16k   | 32k   | 64k   | 128k  | Avg. Accuracy | Avg. Sparsity | Avg. Speedup |
> |---------|-------|-------|-------|-------|-------|-------|--------|---------|-------------|
> |  Baseline (Full Attention)| 95.53 | 92.37 | 92.01 | 87.63 | 84.39 | 76.26 | 88.01    | 0.00             | 1.00x            |
> | MInference    | 95.53 | 92.64 | 91.37 | 85.71 | 83.24 | 67.02 | 85.92    | 0.36 | 0.83x    |
> | DuoAttention  | 95.64 | 92.08 | 90.71 | 84.75 | 83.24 | 75.32 | 86.96| 0.52    | 1.09x|
> | SeerAttention | 95.53 | 92.71 | 92.02 | 88.49 | 83.48 | 73.37 | 87.60| 0.55| 1.41x|
>
> These results demonstrate that SeerAttention achieves the best overall accuracy–efficiency trade‑off, offering higher sparsity and greater speedup without degrading accuracy compared to prior methods.
>
> Reference:
>
> [1] DistServe: Disaggregating Prefill and Decoding for Goodput-optimized Large Language Model Serving, https://www.usenix.org/system/files/osdi24-zhong-yinmin.pdf
>
> [2] Mooncake: A KVCache-centric Disaggregated Architecture for LLM Serving, https://arxiv.org/abs/2407.00079
>
> [3] Quest: Query-aware sparsity for efficient long-context llm inference, https://arxiv.org/abs/2406.10774
>
> [4] H2O: Heavy-Hitter Oracle for Efficient Generative Inference of Large Language Models, https://arxiv.org/abs/2306.14048
>
> [5] Big Bird: Transformers for Longer Sequences, https://arxiv.org/abs/2007.14062
>
> [6] Blockwise Self-Attention for Long Document Understanding, https://arxiv.org/abs/1911.02972
>
> [7] Native Sparse Attention: Hardware-Aligned and Natively Trainable Sparse Attention, https://arxiv.org/abs/2502.11089
>
> [8] MoBA: Mixture of Block Attention for Long-Context LLMs, https://arxiv.org/abs/2502.13189

---

> > ### Comment · Reviewer_ebQ8 · 2025-08-04
> >
> > The authors have improved the paper by adding decoding comparisons between the proposed method and the DuoAttention baseline. SeerAttention demonstrates stronger performance in the decoding-only setup. Although these additional experiments do not fully resolve the comparison issue—namely, that different methods adopt different sparse/dense configurations across prefill and decode—they provide evidence that SeerAttention can perform well under consistent settings.
> >
> > Given these improvements, I am raising my score to borderline accept. However, I strongly encourage the authors to complete the promised experiments, ensure fair and consistent comparisons, and clearly discuss their method's novelty relative to prior works.

---

> ### Author Response · Authors · 2025-08-04
>
> Thank you very much for your helpful feedback and for raising your score. We appreciate your recognition of the improvements of SeerAttention and your suggestions regarding clarity on novelty, which have been valuable in refining the paper. Thanks again for your constructive comments!

---

### Official Review · Reviewer_UEJ9 · 2025-06-30

**Clarity:** 3
**Significance:** 2
**Originality:** 2
**Rating:** 4
**Confidence:** 4

**Summary:**

This paper introduces SeerAttention, a post-training attention mechanism aimed at improving the efficiency of long-context inference in LLMs by learning sparse attention patterns through a self-distilled gating mechanism named AttnGate. Inspired by the gating mechanism in Mixture of Experts (MoE), SeerAttention adds a learnable gate to each attention layer that identifies salient Q-K blocks via pooled Q/K representations. The gating module is trained via KL-divergence to mimic block-level 2D max-pooled full attention outputs, using a lightweight self-distillation setup that freezes the base model. At inference time, a threshold or TopK is applied to the gating outputs to control sparsity.

Empirically, it demonstrates favorable accuracy-speed trade-offs on long-context tasks like LongBench and RULER, with significant speedups (up to 7.3× at 128k sequence length) and minimal accuracy degradation even at high sparsity than existing sparsity based prefilling techniques, such as MInference and DuoAttention.

**Questions:**

1. Comparison to DuoAttention/MInference in Training Cost: How does the training cost of SeerAttention (in GPU hours) compare to  DuoAttention/MInference (lightweight heuristics)? If significantly higher, can the authors provide a cost-benefit analysis?

2. Can ground-truth sparsity (gt) be used directly? In Eq. (2), AttnGate is trained to approximate gt. Why not directly use gt to select top-k blocks (as an accuracy upper bound)? How much accuracy is lost compared to using gt?

3. Is pooling method model-dependent? In section 3.1 and Fig. 2, different pooling strategies yield different performance. Are these consistent across model architectures (e.g., GQA vs MQA heads)? Does the best pooling generalize?

4. Can the authors list and compare tunable parameters required by SeerAttention vs DuoAttention?

5. In Table 1, it is better to show the individual sparsity under different length. For difference context lengths, the sparsity would be difference.

**Ethical Concerns:**

["NO or VERY MINOR ethics concerns only"]

**Final Justification:**

The rebuttal addresses most of my concerns—particularly regarding training cost and training data comparisons with DuoAttention and MInference. These are critical factors for real-world applicability. I also appreciate the additional experimental results on decoding, though I note that the benchmarks used do not focus on long-context input tasks. The proposed method appears to achieve higher sparsity compared to existing sparse techniques, which is promising.

**Limitations:**

1. Training Resource Requirement: Although only the AttnGate is trained, the cost would be higher than heuristic methods. This might limit use in low-resource or deployment-only settings.

2. The paper focuses exclusively on the pre-filling stage and does not discuss how SeerAttention could be extended to the decoding phase. If the proposed gating mechanism or block-sparse attention kernel is not easily compatible with decoding, this could significantly limit the method’s practical applicability, especially in real-time LLM deployments where decoding dominates inference latency.

3. Lack of memory consumption analysis, especially for long context inference.

**Quality:**

3

**Strengths And Weaknesses:**

Strengths:
	1. The proposed method is well-justified and thoughtfully engineered, with a solid motivation tied to dynamic sparsity in attention. The use of self-distillation from full attention as a ground truth is conceptually sound and computationally efficient due to their custom max-pooling kernel.
	2. The work targets a critical bottleneck—long-context inference latency—and achieves consistent improvements across multiple model sizes and benchmarks.
	3. The paper is mostly well-written and thorough.

Weaknesses:
      1. Comparative Cost: The training cost s non-trivial compared to heuristic methods like DuoAttention or MInference. A more detailed comparison of training overhead vs performance gain is missing.
      2. Lack of training data efficiency analysis, the paper relies on the RedPajama dataset for self-distillation but provides no justification for this choice or analysis of how the choice of training data impacts the learned AttnGate. Especially comparing to MInference and DuoAttention. This is important in practical applications.
      3. Dynamic sparsity introduces challenging in batch inference.

---

> ### Author Rebuttal · Authors · 2025-07-29
>
> We thank the reviewer for recognizing that our method is well-justified and thoughtfully designed, especially the motivation around dynamic sparsity in attention, the use of self-distillation with our efficient max-pooling kernel from **both system and algorithm prospective**. This aligns well with our goal to address the long-context prefilling challenge.
>
> ### **1. Training cost vs accuracy gain**
>
> We appreciate your suggestion to clarify the training cost and parameter overhead. In Llama-3-8B-Instruct model,
>
> - SeerAttention: ~40 A100 hours in total (**5 hours on 8×A100**)
>
> - DuoAttention: **“several hours on 8×A100”**
> - MInference: ~2 hours on 1×A100 for sparsity profiling/calibration but is less accurate and slower in inference
>
> SeerAttention requires a training budget comparable to or lower than DuoAttention. Both methods train only the gating module without updating the base model weights, so the cost mainly comes from forward passes. Compared to post-training approaches such as SFT or RL finetuning, this cost is negligible.
>
> For accuracy, the gain should be considered together with sparsity. DuoAttention has a fixed sparsity ratio (~50%), whereas SeerAttention achieves similar accuracy at **much higher sparsity (80–90% on 128k RULER)**. MInference is faster to profile since it only calibrates a sparsity head on a given dataset, but it is less accurate. Moreover, MInference often **suffers from runtime overhead** caused by head-by-head index search and execution flow, as reported in GitHub issues (e.g., https://github.com/microsoft/MInference/issues/43), where users observed that MInference can even run slower than the dense baseline.
>
> As shown in Fig. 9 (Appendix A.1), we design customized training kernels to distill the gate, and the memory/latency overhead during distillation is < 5% compared to native FlashAttention. For LLaMA‑3.1‑8B, the gate adds only 101 M parameters (~1.3% of the model), making the method lightweight and scalable.
>
> | Base Model| AttnGate Size |
> | ---------- | ------ |
> | LLaMA‑3.1‑8B‑Instruct| 101 MB  |
> | LLaMA‑3.1‑70B‑Instruct| 503 MB|
> | Qwen2.5‑7B‑Instruct| 77 MB|
> | Qwen2.5‑14B‑Instruct| 189 MB|
> | Qwen2.5‑32B‑Instruct| 252 MB|
> | DeepSeek‑R1‑Distill‑Qwen‑7B  | 101 MB|
> | DeepSeek‑R1‑Distill‑Qwen‑14B | 189 MB|
> | DeepSeek‑R1‑Distill‑Qwen‑32B | 252 MB|
>
> #### **About training data:**
> Thank you for this question. Selecting training data is less critical in our case as the gate only learns the **distribution instead of the real end-to-end knowledge** from the data. Regarding training data, our gate learns to approximate **distribution patterns rather than full knowledge**, so dataset choice is less critical. We use RedPajama [1] for its popularity in long-context training. MInference and DuoAttention rely on downstream or BookSum [2] datasets, and none of these works provide ablation on dataset choice. This remains an open question but does not undermine our method’s core merit.
>
> ### **2. Use GT directly in inference**
> We appreciate the reviewer’s interest in this question, which was a key consideration in our system–algorithm co-design. Our experiments show that directly using GT to select sparse blocks results in virtually no accuracy loss, even under high sparsity. However, computing GT requires evaluating the full quadratic softmax term, $\text{softmax}(QK^\top)$. If this computation is already performed, there is little performance benefit from only applying sparsity to the $ \text{AttentionScore} \cdot V $ part.
>
> In summary, GT is accurate but expensive, which motivates the need for a fast, low-cost AttnGate.
>
> Regarding the question “How much accuracy is lost compared to using GT?”, intuitively:
>
> - Full attention provides the upper bound for GT-based sparsity.
>
> - GT-based sparsity provides the upper bound for our AttnGate-based sparsity.
>
> As noted above, GT computation is even more expensive than full attention because we must offload all attention scores to CPU to avoid out-of-memory issues before using them as GT. Therefore, we use full attention as the baseline when comparing accuracy across different sparsity settings (see Tables 1–3 in the paper and additional results in the rebuttal to another reviewer: https://openreview.net/forum?id=Nf8yfPDFTl&noteId=yPnYUXTMey).
>
> ### **3. Pooling selection**
> Our pooling experiments are consistent across all tested GQA-based models. The best pooling strategy generalizes well within this setting. However, pooling selection remains an open question, especially for other attention architectures like MQA/MLA. Our main contribution is introducing the **gated attention concept**, predating NSA [3] and MoBA [4] (another reviewer asked). Future work may explore alternative gate structures such as strided MLPs or convolutions, inspired by MoE router literature.
>
> ### **4. Tunable parameters for each layer**
> We would like to clarify to the reviewer that by “tunable parameters” we consider two categories: (1) trainable parameters and (2) hyperparameters. Below, we list and compare these parameters for SeerAttention and DuoAttention accordingly.
>
> #### **For trainable parameters**
>
> - DuoAttention: a scale vector multiplied on streaming head and dense head.
> - SeerAttention: a linear layer from compressed QK to gate hidden vector.
>
> If we compare the number of trainable parameters, DuoAttention indeed has fewer parameters. However, even for SeerAttention, the additional parameters are minimal—less than 1% of the total. For instance, in LLaMA-3.1-8B, the gating module introduces roughly 101M parameters, which is about 101M / 8B ≈ 1.3% of the model size. This overhead is very small and keeps the method efficient to train. As shown in Figure 9 of Appendix A.1, we design customized training kernels to distill the gate, and the memory and latency overhead during this distillation stage are minimal compared to native FlashAttention.
>
> | Base Model| AttnGate Size |
> | ------| ------ |
> | LLaMA‑3.1‑8B‑Instruct| 101 MB|
> | LLaMA‑3.1‑70B‑Instruct| 503 MB|
> | Qwen2.5‑7B‑Instruct| 77 MB|
> | Qwen2.5‑14B‑Instruct| 189 MB|
> | Qwen2.5‑32B‑Instruct| 252 MB|
> | DeepSeek‑R1‑Distill‑Qwen‑7B  | 101 MB|
> | DeepSeek‑R1‑Distill‑Qwen‑14B | 189 MB|
> | DeepSeek‑R1‑Distill‑Qwen‑32B | 252 MB|
>
> #### **For hyperparameters**
>
> SeerAttention introduces an additional hyperparameter, block size, as well as the number of training steps, in addition to the usual hyperparameters required for model fine-tuning. DuoAttention, on the other hand, requires selecting the percentage of streaming heads and full-attention heads, along with the number of training steps and other standard fine-tuning hyperparameters.
>
>
> ### **5. Modify results organization**
> Thank you for the suggestion since we just present the overall sparsity for different sequence length inside the sentences. We will revise the presentation if accepted.
>
> ### **6. About decoding**
> We appreciate the suggestion to discuss decode‑stage evaluation. In this work, we primarily focus on accelerating the prefill stage and leave decode‑stage optimization for future exploration. Prefill and decode are two distinct phases, and several PD‑disaggregation serving systems, such as DistServe [5] and Mooncake [6], have explored separating prefill and decode to improve TTFT/TPOT. Many prior works also focus on only one phase: for instance, MInference mainly targets prefill, while Quest [7] and H2O [8] focus solely on decode.
>
> Our approach can naturally extend to decode sparsity with a minor modification of the AttnGate design. Specifically, we remove the Q‑compression branch (pooling + linear) and instead distill using a 1D max‑pooling ground truth of the attention map (along the sequence dimension) from the base model. In other words, during distillation for decode, we directly use the original Q without sequence‑level compression, enabling each token’s Q to select important KV blocks at inference.
>
> We evaluated this design on the DeepSeek‑R1‑Distill‑Qwen‑14B reasoning model and compared it against Quest. Below we provide a brief summary of the results (we choose block size = 64 for K block). A more complete version can be included in the appendix upon acceptance.
>
> | | Dataset | 2k| 4k | 6k | 8k | Full |
> | --- | --- | --- | --- | --- | --- | --- |
> | SeerAttention-R  | AIME24 | 55.78 | 66.35 | 67.50 | 66.82 | 67.50 |
> |  | MATH500 | 87.65 | 92.10 | 93.05 | 93.12 | 93.30 |
> |  | GPQA-Diamond | 51.26 | 56.79 | 56.41 | 57.48 | 57.80 |
> | Quest  | AIME24 | 25.83 | 46.67 | 53.75 | 60.00 | 67.50 |
> |  | MATH500 | 57.4 | 79.6 | 88.6 | 92.2 | 93.30 |
> |  | GPQA-Diamond | 35.35 | 50.25 | 52.53 | 54.55 | 57.80 |
>
> ### **7.About batch inference**
>
> We appreciate the reviewer’s interest in this question about batching inference. In fact, unlike MoE, batch inference is less challenging for attention. During prefill, **variable-length** inputs can be packed along the sequence dimension, hiding batch size within sequence length. For decoding, modern infrastructures (like vllm or sglang) support **paged attention**, allowing our block-sparse method to selectively activate pages naturally.
>
> Reference:
>
> [1] RedPajama, https://github.com/togethercomputer/RedPajama-Data
>
> [2] BookSum: A Collection of Datasets for Long-form Narrative Summarization, https://arxiv.org/abs/2105.08209
>
> [3] Native Sparse Attention: Hardware-Aligned and Natively Trainable Sparse Attention, https://arxiv.org/abs/2502.11089
>
> [4] MoBA: Mixture of Block Attention for Long-Context LLMs, https://arxiv.org/abs/2502.13189
>
> [5] DistServe: Disaggregating Prefill and Decoding for Goodput-optimized Large Language Model Serving, https://www.usenix.org/system/files/osdi24-zhong-yinmin.pdf
>
> [6] Mooncake: A KVCache-centric Disaggregated Architecture for LLM Serving, https://arxiv.org/abs/2407.00079
>
> [7] Quest: Query-aware sparsity for efficient long-context llm inference, https://arxiv.org/abs/2406.10774
>
> [8] H2O: Heavy-Hitter Oracle for Efficient Generative Inference of Large Language Models, https://arxiv.org/abs/2306.14048

---

> > ### Comment · Reviewer_UEJ9 · 2025-08-07
> >
> > Thanks for the detailed response, which addresses most of my concerns—particularly regarding training cost and training data comparisons with DuoAttention and MInference. These are critical factors for real-world applicability. I also appreciate the additional experimental results on decoding, though I note that the benchmarks used do not focus on long-context input tasks.
> >
> > The proposed method appears to achieve higher sparsity compared to existing sparse techniques, which is promising. I am inclined to raise my score to a borderline accept. I would still encourage the authors to further emphasize the paper’s strengths in terms of training cost and its dependence on training data. Additionally, it would be helpful to clarify whether the method is compatible with well-known decoding speedup techniques—particularly KV eviction methods, which are often designed for full-attention models rather than sparse attention.

---

> > > ### Author Response · Authors · 2025-08-07
> > >
> > > Thank you for the thoughtful feedback. We're glad our response (like the comparisons on training cost and data efficiency) helped clarify things. We'll make sure to emphasize these strengths more clearly in the revised paper. We also appreciate the point about KV cache eviction/compression, and will include a discussion on its compatibility with sparse attention and the potential trade-offs involved. Thanks again for your helpful suggestions!

---

### Official Review · Reviewer_DFJm · 2025-07-01

**Clarity:** 2
**Significance:** 3
**Originality:** 4
**Rating:** 5
**Confidence:** 4

**Summary:**

The paper introduces SeerAttention—a post-training sparse attention method that augments the attention block with a learnable gating layer, which can identify block-sparsity patterns in attention maps. The gating layer introduced in the paper adds minimal overhead. It does not require from-scratch training, and the authors train it using a self-distillation technique, employing a KL-divergence loss with the ground-truth attention sparsity. They also augment the flash-attention kernel with their pooling scheme to ensure efficient attention during training of the gating layers. Following a block-sparsity pattern, the authors ensure compatibility with the FlashAttention kernel, making SeerAttention practically useful and demonstrating speedup over FA over large sequence lengths. The authors show extensive evaluation over several models and benchmarks and compare against various baselines. Their results show promising results with SeerAttention outperforming SOTA sparsity methods while being more efficient.

**Questions:**

- Can the authors show the results for LongBench and RULER with different sparsity levels for SeerAttention?
- In the Short-Context evaluation, MMLU and ARC-c have very low average sparsities - 3.4 and 26. This is not the case for other short-context benchmarks. Have the authors analysed why that is the case? Is it possible that, depending on the dataset used to train the gate layers, it may not detect sparsity in certain downstream tasks?
- What is the memory overhead of gate parameters added to the model?
- SeerAttention applies block PE to the non-RoPE Q and K. As far as I know, the keys in KV-Cache are stored after RoPE application. How does SeerAttention handle this?

**Ethical Concerns:**

["NO or VERY MINOR ethics concerns only"]

**Final Justification:**

I maintain that SeerAttention is a novel, sound and practically relevant sparse attention technique that outperforms SOTA techniques like DuoAttention. The method is extensively evaluated, well presented and practically efficient. My major concerns were as follows:

- Evaluation on wider sparsity values: The authors provide data in the rebuttal to show that SeerAttention performs well even in higher sparsity regimes. They also offer accuracy vs speed tradeoff evaluations for different sparsity levels.
- Cost of training the gating layers: While this remains a limitation of the method, the authors have shown that in comparison to other SOTA methods like DuoAttention, this cost is the same or less for SeerAttention.
- Applicability to Decode Settings: While the paper focuses on prefill-only acceleration, which is a completely valid and important setting, the authors also provide results on how their method can be modified to work with decode. This is impressive and enhances the paper's contribution.

In summary, all of my major concerns have been addressed and my questions answered. Therefore, I maintain my score of accept.

**Limitations:**

Limited discussion. While the authors do mention future work in their conclusion with potential for improvements to SeerAttention, having a dedicated section on Limitations would be helpful. There are several limitations which can be discussed: applicability to larger models due to post-training requirements, applicability to the decode stage in inference, and memory overhead due to added gating parameters.

**Quality:**

3

**Strengths And Weaknesses:**

## Strengths
- The information in the paper is very well presented. The authors describe their evaluation setup in detail and explain the choices made in their design and training of SeerAttention fairly well.
- SeerAttention is compatible with FlashAttention, and shows practical speedup over the SOTA attention method for long context inference. This makes it deployable in practical settings.
- The evaluation is extensive, with SeerAttention showing minimal drop in accuracy over multiple models and benchmarks. The attention maps identified by the gate layer also look promising.

## Weaknesses
- The choice of threshold in the evaluations is such that the sparsity levels for SeerAttention for both RULER and LongBench hover around 0.5-0.6. These are fairly low sparsity regimes, and attention maps have much higher sparsity. It would be great if the authors could show benchmark accuracy results for a wider range of sparsity with SeerAttention
- The authors mention show ~7x speedup using SeerAttention, but that occurs at 90% sparsity, and there is no information in the paper to understand the accuracy trade-off with that kind of sparsity. At 50-60% sparsity, the speedups are more in the range of 1.5-1.6, which can be considered good.
- While the training scheme for the gate layers developed by the authors is efficient and does not require full model retraining, it is still expensive. By the author’s own experiments, training the gate layers of a 7B model can take up to 40 A100 hours. For larger models like Llama 70B or 405B, this might be even more expensive. In this regime, this might be a big trade-off when compared to techniques that require no post-training.
- SeerAttention only optimises the pre-fill stage and thus will not provide much benefit in decode-heavy workloads

---

> ### Author Rebuttal · Authors · 2025-07-29
>
> We thank the reviewer for their thoughtful feedback and for recognizing the novelty.
>
> ### **1. Weaknesses 1 & 2: Sparsity Ratio and Speedup**
> We evaluated speedup–accuracy trade‑offs under varying sparsity. The table below shows kernel‑level speedups for different sparsity and sequence lengths:
>
> | Sparsity / SeqLen | 8k   | 16k  | 32k  | 64k  | 128k |
> |-------------------|------|------|------|------|-------|
> | 50%               | 1.73x| 1.78x| 1.81x| 1.84x| 1.84x |
> | 70%               | 2.58x| 2.80x| 2.94x| 3.01x| 3.01x |
> | 90%               | 5.91x| 6.93x| 7.87x| 8.28x| 8.63x |
>
> In the end-to-end, by adjusting threshold T, SeerAttention flexibly balances sparsity and accuracy. RULER‑128k accuracy results:
>
> | Method                           | Sparsity | Accuracy (%) |
> | -------------------------------- | -------- | ------------ |
> | DuoAttention 0.5| 0.50     | 75.32 |
> | DuoAttention 0.7| 0.70     | 70.35        |
> | DuoAttention 0.75| 0.75     | 55.48        |
> | SeerAttention $T=5\times10^{-5}$ | 0.57  | 75.71  |
> | SeerAttention $T=1\times10^{-4}$ | 0.62     | 75.15 |
> | SeerAttention $T=2\times10^{-4}$ | 0.72     | 74.94|
> | SeerAttention $T=4\times10^{-4}$ | 0.80     | 73.81  |
> | SeerAttention $T=5\times10^{-4}$ | 0.83   | 73.37  |
>
>
> ### **Tradeoffs between Speedup and Accuracy**
>
> To evaluate the tradeoffs, we count the average sparsity and accuracy to present.In LongBench, SeerAttention employs a threshold of 2e-3 for all AttnGates. With the same threshold, different attention gates can exhibit varying sparsity ratios, and longer context data tends to be sparser.
>
>
> #### **LongBench performance & efficiency results on Llama-3.1-8B-Instruct**
>
> | Methods| 0-4k  | 4-8k  | 8k+   | Avg. Accuracy | Avg. Sparsity | Avg. Speedup |
> |----------|-------|-------|-------|-----------|---------------|---------|
> |  Baseline (Full Attention)  | 55.32 | 53.98 | 52.90 | 54.07| 0.00  | 1.00x    |
> | MInference| 55.23 | 53.78 | 52.18 | 53.73     | 0.31     | 0.72x|
> | MoA| 50.74 | 49.84 | 51.89 | 50.82     | 0.35| 1.11x|
> | DuoAttention    | 53.77 | 52.17 | 51.27 | 52.40     | 0.50*   | 1.07x|
> | SeerAttention | 55.43 | 54.49 | 52.69 | 54.20 | 0.50 | 1.32x |
>
> \* 50% streaming heads, the real sparsity < 50%
>
> In RULER, we employ a threshold of 5e-4 under 4k-128k test benches.
>
> #### **RULER performance & efficiency results on Llama-3.1-8B-Instruct**
>
> | Methods       | 4k    | 8k    | 16k   | 32k   | 64k   | 128k  | Avg. Accuracy | Avg. Sparsity | Avg. Speedup |
> |---------|-------|-------|-------|-------|-------|-------|--------|---------|-------------|
> |  Baseline (Full Attention)| 95.53 | 92.37 | 92.01 | 87.63 | 84.39 | 76.26 | 88.01    | 0.00             | 1.00x            |
> | MInference    | 95.53 | 92.64 | 91.37 | 85.71 | 83.24 | 67.02 | 85.92    | 0.36 | 0.83x    |
> | DuoAttention  | 95.64 | 92.08 | 90.71 | 84.75 | 83.24 | 75.32 | 86.96| 0.52    | 1.09x|
> | SeerAttention | 95.53 | 92.71 | 92.02 | 88.49 | 83.48 | 73.37 | 87.60| 0.55| 1.41x|
>
> These results demonstrate that SeerAttention achieves the best overall accuracy–efficiency trade‑off, offering higher sparsity and greater speedup without degrading accuracy compared to prior methods.
>
> ### **2. Weakness 3 & Question 3: Training Cost and Parameter Overhead**
> We appreciate your suggestion to clarify the training cost and parameter overhead. In Llama-3-8B-Instruct model,
>
> - SeerAttention: ~40 A100 hours in total (**5 hours on 8×A100**)
>
> - DuoAttention: **“several hours on 8×A100”**
> - MInference: ~2 hours on 1×A100 for sparsity profiling/calibration but is less accurate and slower in inference
>
> The training cost of SeerAttention is comparable to or lower than DuoAttention. Although MInference calibrates faster, it sacrifices inference accuracy and speed. SeerAttention introduces only a small number of trainable parameters. As shown in Fig. 9 (Appendix A.1), we design customized training kernels to distill the gate, and the memory/latency overhead during distillation is < 5% compared to native FlashAttention. For LLaMA‑3.1‑8B, the gate adds only 101 M parameters (~1.3% of the model), making the method lightweight and scalable.
> | Base Model                   | AttnGate Size |
> | ---------------------------- | ------------- |
> | LLaMA‑3.1‑8B‑Instruct| 101 MB  |
> | LLaMA‑3.1‑70B‑Instruct| 503 MB        |
> | Qwen2.5‑7B‑Instruct| 77 MB         |
> | Qwen2.5‑14B‑Instruct| 189 MB|
> | Qwen2.5‑32B‑Instruct| 252 MB        |
> | DeepSeek‑R1‑Distill‑Qwen‑7B  | 101 MB|
> | DeepSeek‑R1‑Distill‑Qwen‑14B | 189 MB        |
> | DeepSeek‑R1‑Distill‑Qwen‑32B | 252 MB        |
>
>
> ### **3. Weakness 4: Decoding-Stage Inference:**
>
> We appreciate the suggestion to discuss decode‑stage evaluation. In this work, we primarily focus on accelerating the prefill stage and leave decode‑stage optimization for future exploration. Prefill and decode are two distinct phases, and several PD‑disaggregation serving systems, such as DistServe [1] and Mooncake [2], have explored separating prefill and decode to improve TTFT/TPOT. Many prior works also focus on only one phase: for instance, MInference mainly targets prefill, while Quest [3] and H2O [4] focus solely on decode.
>
> Our approach can naturally extend to decode sparsity with a minor modification of the AttnGate design. Specifically, we remove the Q‑compression branch (pooling + linear) and instead distill using a 1D max‑pooling ground truth of the attention map (along the sequence dimension) from the base model. In other words, during distillation for decode, we directly use the original Q without sequence‑level compression, enabling each token’s Q to select important KV blocks at inference.
>
> We evaluated this design on the DeepSeek‑R1‑Distill‑Qwen‑14B reasoning model using two challenging math tasks and compared it against Quest. Below we provide a brief summary of the results (block size = 64). A more complete version can be included in the appendix upon acceptance. The results show that SeerAttention achieves higher accuracy than Quest, even with larger KV blocks and sparser attention.
>
> #### **Model: DeepSeek-R1-Distill-Qwen14B**
>
> | Method (**AIME**)                                       | Acc   | Avg Sparsity | Avg Decode Length |
> |----------------------------|-------|--------------|-------------------|
> | Full Attention                                   | 70    | 0            | 10k               |
> | Quest (Block Size 16, Token Budget 8192)         | 53.33 | 36%          | 17k               |
> | Quest (Block Size 64, Token Budget 8192)         | 16.67 | 65%          | 28k               |
> | SeerAttention (Block Size 64, Threshold=5e-3)    | 66.7  | 86%          | 11k               |
>
>
> | Method (**MATH500**)                                          | Acc   | Avg Sparsity | Avg Decode Length |
> |--------------------------------------------------|-------|--------------|-------------------|
> | Full Attention                                   | 94    | 0            | 4k                |
> | Quest (Block Size 16, Token Budget 2048)         | 83.4  | 73%          | 6k|
> | SeerAttention (Block Size 64, Threshold=7e-3)    | 94    | 85%          | 4k|
> | SeerAttention (Block Size 64, Threshold=1e-2)    | 92.8  | 89%          | 4k|
>
>
> ### **4. Question 2: Short Context:**
> We appreciate the suggestion to clarify why certain short‑context tasks (e.g., MMLU, ARC‑c) exhibit lower sparsity.
> First, as the input sequence length decreases, the attention map — Softmax(QKᵀ) — naturally becomes denser: shorter sequences lead to more concentrated information, and the softmax distribution assigns higher average attention scores to each token. This behavior is inherent to attention computation itself and not specific to our gate predictor design.
> In SeerAttention, we can freely control sparsity by selecting different thresholds or Top‑K ratios. For a fixed threshold, the resulting sparsity naturally decreases for short contexts and increases for long contexts because the gating mechanism’s softmax rows sum to 1, so tokens in shorter sequences inherently receive larger attention scores.
> Moreover, attention computation is rarely the bottleneck for short contexts, whereas sparse attention provides significant efficiency gains for long‑context inference. Attention only becomes a major bottleneck when the sequence is long, where sparsity leads to meaningful speedups.
> We thank the reviewer for pointing this out — we have reorganized Table 3 accordingly. As sequence length decreases, the observed sparsity drops (e.g., 3.4% and 26%), which directly reflects this inherent behavior.
>
>
> |       | MMLU | ARC-c | HellaS. | GSM-8K |
> |-------|------|-------|---------|-----|
> | Full Attention| 68.1 | 60.7  | 80.1| 75.7   |
> | SeerAttention| 67.9 | 60.2  | 79.8| 75.6   |
> | Avg Sparsity| 3.4  | 26| 50.4    | 52.1   |
> | Avg SeqLens| 118  | 395   | 840| 872    |
>
>
> ### **5. Question 4: Rope in KV cache**
> The rope design in decoding version is similar to prefill case. In decoding, SeerAttention maintains a small compressed K cache (after RoPE) specifically for gating. This avoids loading the full K cache at every step and allows selective V-loading, maximizing decode-stage speedup. Without this, sparsity-based selection would be limited to a **~2× upper-bound** speedup due to full K-cache loading.
>
> Once again, we sincerely thank the reviewer for their positive assessment of SeerAttention’s **novelty, technical soundness, and practical impact**. We will incorporate your suggestions to further strengthen the final version of the paper.
>
> [1] DistServe: Disaggregating Prefill and Decoding for Goodput-optimized Large Language Model Serving, https://www.usenix.org/system/files/osdi24-zhong-yinmin.pdf
>
> [2] Mooncake: A KVCache-centric Disaggregated Architecture for LLM Serving, https://arxiv.org/abs/2407.00079
>
> [3] Quest: Query-aware sparsity for efficient long-context llm inference, https://arxiv.org/abs/2406.10774
>
> [4] H2O: Heavy-Hitter Oracle for Efficient Generative Inference of Large Language Models, https://arxiv.org/abs/2306.14048

---

> > ### Comment · Reviewer_DFJm · 2025-08-04
> >
> > I thank the authors for their response and appreciate the detailed response to my concerns and questions. I maintain that SeerAttention is a novel, sound and practically relevant sparse attention technique that outperforms SOTA techniques like DuoAttention. The method has a few limitations, like: (1) High training cost for the gating layers, (2) Careful selection of thresholds to achieve the desired sparsity, and (3) limited applicability to the decode stage [minor] (the rebuttal response compares with Quest but the sparsity levels are too different to make meaningful inferences and I understand this is not the main focus of the paper). Despite these, I believe the technique is practical, efficient, well-evaluated and marks a good improvement over existing methods.
> > With these considerations in mind, I maintain my original score of accept.

---

> ### Author Response · Authors · 2025-08-04
>
> Thank you very much for your continued support and for highlighting the novelty and practical relevance of SeerAttention! We’re glad to hear that you find the method sound and well-evaluated.
>
> Regarding point (2) on the careful selection of thresholds—we’d like to clarify that SeerAttention also supports TopK-based masking, which enables precise control over the sparsity level. This is shown in the right plot of Figure 1. While we chose threshold-based evaluation in the main experiments due to its ability to dynamically adapt sparsity across different sequence lengths without notable performance loss, users can also specify a desired sparsity level directly through TopK during inference.
>
> We sincerely appreciate your thoughtful feedback and your recommendation!

---

### Note · Authors · 2025-08-14

We sincerely appreciate all reviewers’ and AC’s efforts for the insightful and constructive comments. We are glad that the reviewers recognized the following strengths:

 - **Novelty and Methodology**: SeerAttention is a novel, sound, and practically relevant sparse attention technique that outperforms SOTA methods such as DuoAttention (Reviewer DFJm), introducing a learnable block-wise attention gating mechanism. Reviewer HW7W increased the score on novelty after our clarification of originality.

- **Performance and Efficiency**: SeerAttention achieves impressive efficiency improvements (Reviewer DFJm), with optimized memory usage and lightweight overhead (Reviewer UEJ9). Reviewers UEJ9 and ebQ8 valued its customized training and inference CUDA/Triton kernels, compatibility with native FlashAttention for long-context inference.

- **Experimental Rigor**: Comprehensive ablation studies and evaluations on various benchmarks demonstrate SeerAttention’s robust performance. Reviewers DFJm, UEJ9, and HW7W recognized the breadth of evaluations across multiple models and benchmarks, including kernel-level and e2e performance, efficiency and accuracy tradeoffs, decoding-stage adaptation, and block-size ablations.

 - **Presentation and Clarity**: Our paper is well-structured, with thorough, clear writing and strong motivation (Reviewers UEJ9 and DFJm).

In the rebuttal, we further addressed concerns by providing: (1) detailed efficiency and accuracy tradeoffs confirming SeerAttention’s advantages across regimes; (2) clarifications on training cost and minimal memory overhead; (3) new decoding-stage results outperforming Quest on reasoning tasks; and (4) block-size ablations and additional novelty clarifications.

After the discussion, we are grateful for the reviewers’ recognition—DFJm maintained accept, UEJ9 and ebQ8 raised their ratings to borderline accept, and HW7W, whose concern about novelty was addressed, increased the score from borderline reject to (borderline) accept—confirming the consensus that SeerAttention is novel, efficient, and technically sound for long-context LLM inference.

These acknowledgments highlight the value of SeerAttention in addressing critical challenges in long-context LLM inference. We are committed to refining our work based on the constructive feedback provided. Thank you for your time and thoughtful suggestions.

Best regards,

Authors

---

### Decision · Program_Chairs · 2025-09-17

**Decision:**

Accept (poster)

**Comment:**

This paper presents SeerAttention, a dynamic block-sparse attention mechanism that learns block-level sparsity for the prefilling stage of large language models. Rather than relying on fixed sparsity patterns, SeerAttention uses a lightweight gating module (inspired by MoE) that pools and projects Q/K blocks to produce gating scores and selectively activate important attention blocks. Integrated with a block-sparse FlashAttention kernel and trained via self-distillation (KL alignment to dense masks), the method substantially reduces latency while maintaining or improving model accuracy.

Reviewers agree that SeerAttention is novel, technically sound, and practically relevant. The learnable block-wise gating distinguishes the approach from prior heuristics, yielding meaningful efficiency and memory improvements with modest overhead. The paper provides thorough empirical evaluation—kernel-level and end-to-end benchmarks, ablations (including block-size and training/decoding tradeoffs), and decoding-stage results—that convincingly demonstrate favorable accuracy–efficiency tradeoffs. The presentation is clear and well-structured. Overall, SeerAttention makes a noteworthy contribution to efficient long-context LLM inference and is judged by reviewers to be both methodologically innovative and empirically strong. I recommend acceptance, subject to minor revision to ensure reproducibility (e.g., releasing kernels/integration details and clarifying training costs and overheads).